# FreqPolicy: Frequency Autoregressive Visuomotor Policy with Continuous Tokens

**Yiming Zhong**[1], **Yumeng Liu**[2], **Chuyang Xiao**[1], **Zemin Yang**[1],
**Youzhuo Wang**[1], **Yufei Zhu**[1], **Ye Shi**[1], **Yujing Sun**[2,3], **Xinge Zhu**[4], **Yuexin Ma**[1*]
[1]ShanghaiTech University [2]The University of Hong Kong
[3]Digital Trust Centre, Nanyang Technological University[4]The Chinese University of Hong Kong
{zhongym2024, mayuexin}@shanghaitech.edu.cn
**Project Page:** https://freq-policy.github.io/
**Code:** https://github.com/4DVLab/Freqpolicy

## Abstract

Learning effective visuomotor policies for robotic manipulation is challenging, as it requires generating precise actions while maintaining computational efficiency. Existing methods remain unsatisfactory due to inherent limitations in the essential action representation and the basic network architectures. We observe that representing actions in the frequency domain captures the structured nature of motion more effectively: low-frequency components reflect global movement patterns, while high-frequency components encode fine local details. Additionally, robotic manipulation tasks of varying complexity demand different levels of modeling precision across these frequency bands. Motivated by this, we propose a novel paradigm for visuomotor policy learning that progressively models hierarchical frequency components. To further enhance precision, we introduce continuous latent representations that maintain smoothness and continuity in the action space. Extensive experiments across diverse 2D and 3D robotic manipulation benchmarks demonstrate that our approach outperforms existing methods in both accuracy and efficiency, showcasing the potential of a frequency-domain autoregressive framework with continuous tokens for generalized robotic manipulation.

## 1 Introduction

The study of visuomotor policies enable robots to learn task execution from demonstrations by leveraging raw visual inputs, such as images or point clouds, allowing them to generate effective action sequences in response to new visual observations. It has become a prevailing paradigm in robot manipulation [10]. However the requirement of high precision in robotic tasks and the sequential correlation in action space present challenges for visuomotor policy learning.

Existing methods for visuomotor policy learning can be broadly categorized into diffusion-based methods [10, 48, 38, 42, 47, 15, 43, 39] and autoregressive (AR) methods [13, 16, 49]. Diffusion-based approaches model the action distribution conditioned on observations, leveraging their powerful generative capabilities to produce reliable and diverse action sequences even from limited demonstrations. However, diffusion models typically encounter higher computational costs and increased inference latency due to their iterative sampling process, which could limit their practicality in efficiency-sensitive applications. In contrast, AR methods sequentially predict each action step conditioned on previous actions and current observations. They are generally more computationally efficient and enable faster inference, making them attractive for real-time control scenarios. Nevertheless, AR approaches may be prone to compounding errors over long horizons, and due to their common reliance on discrete representations, often struggle to accurately model inherently continuous action spaces, limiting their ability to capture complex temporal correlation. Despite

---

[*]Corresponding author.

39th Conference on Neural Information Processing Systems (NeurIPS 2025).

recent advances, both diffusion-based and AR methods share a fundamental limitation: they overlook the diversity of the action space arising from task complexity and the degrees of freedom in robotic manipulation. Representing the inherent structured features of different actions is of great significance for the generalization and robustness of visuomotor policies.

To address these limitations, we rethink the robotic action representation from a frequency-domain perspective. Figure 1 shows one example of the action signals (from Adroit Door [30]) reconstructed by filtering different frequency bands. We can see that the first $30\%$ of frequency bands is already sufficient to allows us to reconstruct a signal nearly identical to the original, preserving $95\%$ of its energy. Retaining just the first $10\%$ of bands can recover the general trend of the signal. Incorporating high-frequency information, up to the first $60\%$ of frequency bands, enables the restoration of finer details, such as subtle oscillations. Meanwhile, through extensive statistical analysis of robot task execution behaviors, we found the required frequency components vary depending on the complexity of different tasks. For simple tasks, such as pick-and-place, low-frequency information is often sufficient, and high-frequency signals may be redundant, especially the collection of demonstrations in real world

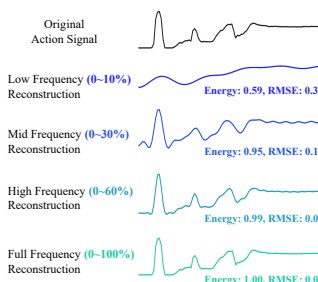

Figure 1: Action signals reconstructed across different frequency bands; *see Appendix for energy details.*

can introduce noise or unnatural jitter. Filtering out these components results in smoother, more natural actions. However, for complex tasks such as dexterous manipulation, high-frequency details are essential for precise control. In summary, for continuous action spaces, investigating hierarchical frequency-domain information and developing effective representation and modeling methods for spectral features are crucial for enabling robots to perform tasks with varying complexity levels.

With this observation, we propose to learn the visuomotor policies based on the modeling of hierarchical frequency domains. Notably, low-frequency information is easier to learn and captures the global structure of the motion. Rather than producing the entire frequency spectrum at once, we adopt a multi-stage progressive approach, starting from low-frequency representations and gradually extending to full-spectrum actions. This corse-to-fine generation process not only simplifies the modeling complexity of sophisticated actions but also provides low-frequency actions with protection from high-frequency noise, ensuring stability in the fundamental motion structure. Autoregressive (AR) paradigm, with their sequential modeling capabilities, are inherently well-suited to implement this coarse-to-fine generation paradigm, effectively capturing hierarchical dependencies between frequency domains. However, previous AR-based methods [29] usually discretize the inherently continuous action space, resulting in significant information loss. Recent study [19] also suggests that discretization of originally continuous spaces is not necessary for AR modeling, and diffusion models can better representing continuous probability distributions. Motivated by these insights, we propose a coarse-to-fine robotic action generation paradigm, **FreqPolicy**, that combines continuous action representations with AR modeling in the frequency domain.

To be specific, our approach utilizes the Discrete Cosine Transform (DCT) [18] to convert action sequences into frequency components. Leveraging a masked encoder-decoder architecture, we map trajectories from various frequency bands into distinct latent codes. It predicts refined motions progressively, where low-frequency signals guide the generation of high-frequency details. Our framework bridges the gap between hierarchical frequency representations and probabilistic modeling, offering a unified and scalable solution for visuomotor policy learning. Extensive experiments show that our method yields significant improvements on challenging robotic manipulation benchmarks, demonstrating both its efficiency and state-of-the-art performance for generating precise and high-fidelity actions. In summary, our contributions are as follows:

- We have explored frequency-based action space representation and propose a novel solution for effective visuomotor policy learning by progressively modeling hierarchical frequency components to capture the structured nature of robotic motions.

- We introduce continuous latent representations with diffusion-based decoding, eliminating discretization requirements while preserving action space continuity and autoregressive efficiency.

- Our method achieves state-of-the-art performance on extensive robotic benchmarks, significantly outperforming others in both success rate and computational efficiency.

## 2 Related Work

### 2.1 Action Representations for robots

Action representation is important in robotic learning, as it determines how agents encode and generate behaviors in complex environments. Recently, transformers have achieved remarkable success in large language models (LLMs)[8, 1, 36], demonstrating exceptional versatility in sequence generation tasks. At the same time, diffusion models have shown strong generative capabilities for images[17, 44] and have garnered significant attention for their adaptation to robotic policies [10, 48]. The advancements in transformer and diffusion models have given rise to two primary paradigms for action representation: discrete and continuous. Transformer models typically employ discrete tokenization, encoding actions as sequences of tokens. In robotic tasks, discretization can be realized through straightforward quantization of each action dimension at every timestep [7, 6] or clustering approaches such as BeT [32], which enable the generation of diverse behaviors but may sacrifice fine-grained control.

In contrast, diffusion models and other continuous approaches [33, 41, 10, 48, 50, 14] leverage probabilistic frameworks—such as VAEs, diffusion processes, and normalizing flows—to model the action space directly. These methods preserve the full expressiveness and precision of the original action space, avoiding the loss of nuance that can result from discretization. However, this advantage often comes at the cost of increased computational complexity, particularly in high-dimensional settings.

Recently, frequency-domain strategies have been explored to address redundancy in both images [28, 20, 45] and action spaces. FAST [29] demonstrates that excessive high-frequency information in action sequences can hinder model training, and propose a compression-based tokenization scheme that reduce redundancy in action signals and improves training efficiency. However, both frequency-domain compression and action discretization inevitably cause information loss, which limit fine-grained, precise synthesis. In this work, we combine frequency-domain action representation with continuous sequence modeling. Our approach enables flexible transitions between spatial and frequency domains, preserving critical details for high-fidelity motion while maintaining efficiency, thus supporting more expressive and scalable robotic policy learning.

### 2.2 Visuomotor Policy for Robotic Manipulation

Visuomotor policies for robotic manipulation map visual inputs directly to control actions, enabling robots to interact with their environments in a closed feedback loop. Broadly, there are two main approaches to tackling this problem: one is to use diffusion based methods, and the other is to employ autoregressive methods. Diffusion-based methods typically generate action segments by modeling the conditional distribution of actions given observations using diffusion models [10, 48, 38, 42]. Autoregressive methods generate action sequences step-by-step, predicting each action based on previous outputs [21, 13, 49, 40]. While this approach is efficient, it often lacks long-horizon structural modeling. Recently, several coarse-to-fine methods have been proposed, but they primarily operate in the temporal domain. CARP [16] incorporates multi-scale reconstruction from VAR [35] for action generation, but depends on discrete representations derived from VQ-VAE [37], which limits precision. DensePolicy [34] introduces a bidirectional, BERT-inspired [11] framework for hierarchical, coarse-to-fine action prediction. However, its iterative refinement relies on progressively increasing temporal upsampling density, which does not remove high-frequency noise or compress redundant information in action signals. Unlike previous methods, our approach performs AR generation in the frequency domain, modeling signals at different frequencies independently. This separation prevents low-frequency components from being affected by high-frequency noise. Moreover, for high-dimensional and complex tasks, progressively learning high-frequency signals from low-frequency components eases the challenge of direct generation.

## 3 Method

### 3.1 Overview

**Problem Formulation** Given a dataset of paired sequences $\mathcal{D} = \{(\mathbf{o}, \mathbf{x})\}$, where each $\mathbf{o} = [o_0, \ldots, o_{N-1}]^T$ denotes a sequence of observations (e.g., RGB images, depth maps, or point clouds)

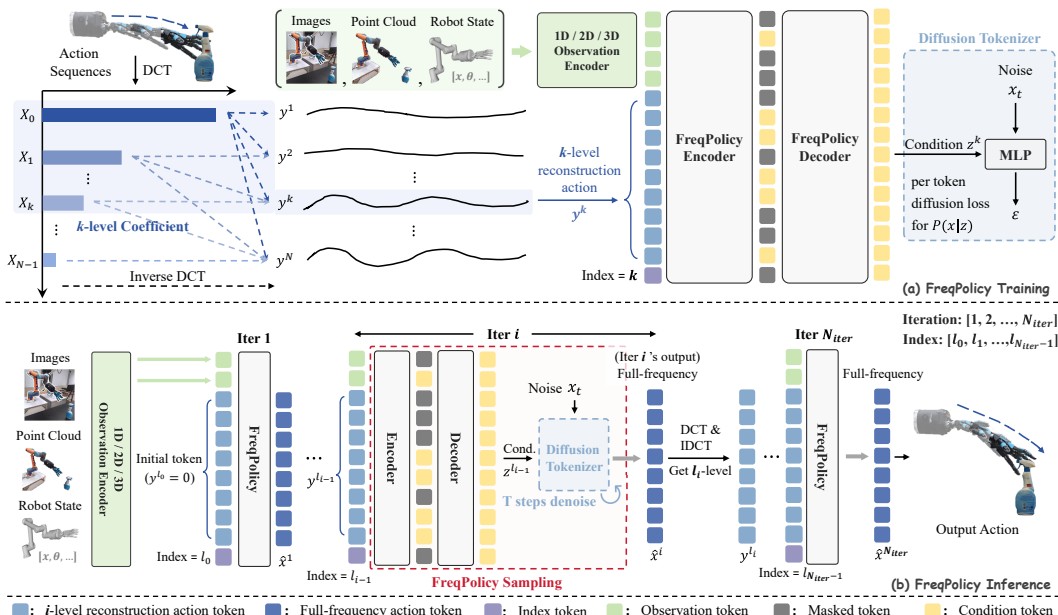

Figure 2: **Pipeline of FreqPolicy showing both training (a) and inference (b) procedures.** We first transform action trajectories into the frequency domain via DCT, and then learns latent codes for different frequency level actions using FreqPolicy, and reconstructs actions through masked prediction and a diffusion-based decoder. This enables robust, frequency-aware, and high-fidelity robotic action generation.

and each $\mathbf{x} = [x_0, x_1, \ldots, x_{N-1}]^T \in \mathbb{R}^{N \times d}$ represents the corresponding sequence of robot actions with variable length $N$ and action dimension $d$, our objective is to train a policy that can produce action trajectories $\hat{\mathbf{x}}$ based on new observation sequences $\hat{\mathbf{o}}$ to effectively perform the demonstrated tasks.

As shown in Figure 2, our method consists of two stages. **During training (Figure 2a)**, we apply DCT to decompose action sequences into frequency components and reconstruct them at different levels via inverse DCT. For frequency level $k$, the FreqPolicy encoder-decoder processes the $k$-level reconstruction $\mathbf{y}^k$, observation features, and frequency index to produce a continuous token $\mathbf{z}^k$, which conditions a diffusion model to reconstruct the original full-frequency action $\mathbf{x}$. We adopt frequency-aware masking (gray blocks in the figure) to improve training efficiency. **During inference (Figure 2b)**, we perform hierarchical generation through $N_{\text{iter}}$ iterations. Starting from zero ($\mathbf{y}^{l_0} = \mathbf{0}$), each iteration $i$ generates a full-frequency action $\hat{\mathbf{x}}^i$ conditioned on the previous output $\mathbf{y}^{l_{i-1}}$, then filters it to $\mathbf{y}^{l_i}$ for the next iteration. This coarse-to-fine process progressively refines actions from low-frequency global structure to high-frequency fine details.

This section is organized as follows: Section 3.2 analyzes action trajectories in the frequency domain; Section 3.3 introduces the space-frequency transformation method employed in our approach; Section 3.4 describes the continuous token representation with diffusion models; and Section 3.5 presents the frequency-based hierarchical generation strategy.

## 3.2 Frequency Domain Analysis

To motivate our approach, we first conduct a systematic analysis of action trajectories under various task conditions. We selected two representative benchmarks to evaluate our approach. For complex manipulation tasks, we chose three tasks from the Adroit Benchmark [30], which utilizes a dexterous robotic hand with 26 degrees of freedom. For simpler tasks, we selected three tasks from the Robomimic Benchmark [23], which features parallel grippers with only 10 degrees of freedom. On these tasks, we aim to analyze the energy distribution of each action dimension of the robotic arm across different frequency bands, as well as to investigate the impact of discarding high-frequency signals on task performance.

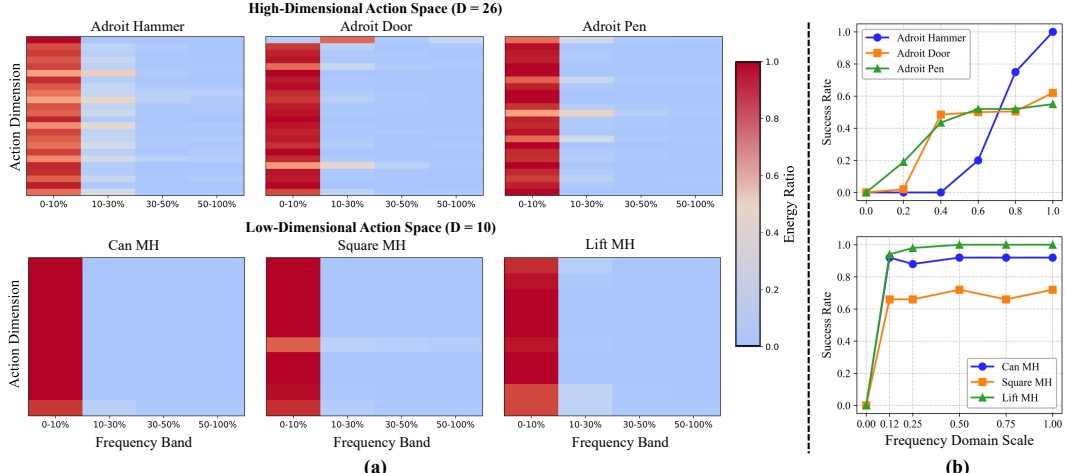

Figure 3: *(a)* **Heat maps of frequency band energy across action dimensions for different tasks.** The top row shows Adroit [30] tasks with high-dimensional actions (26 dimensions), while the bottom row presents Robomimic [23] tasks with low-dimensional actions (10 dimensions). *(b)* **Success rate of actions reconstructed with varying frequency ratios.** We reconstruct action sequences using different proportions of frequency components and evaluate their success rates on the original tasks.

**Energy Distribution.** We present the energy heat maps for these tasks in Figure 3*(a)*, where the vertical axis represents different dimensions of the action space, corresponding to the joint angles of the robotic hand or arm, and the horizontal axis indicates the energy ratio across different frequency bands. The red regions indicate areas with a high energy ratio, suggesting that these frequency bands contain a greater proportion of the signal's energy. Conversely, the blue regions represent areas with a low energy ratio. In both the Adroit tasks (top row) and the Robomimic tasks (bottom row), most of the energy is concentrated in the 0–10% low-frequency bands (leftmost columns), which is why these regions appear red. By comparing the heat map for tasks in different action space, we found that tasks with high-dimensional actions show greater variance in the energy distribution compared to low-dimensional tasks. Specifically, the energy distribution of the action space in the Adroit Benchmark shows distinct low-frequency (0-10%) patterns across its three tasks, while in the Robomimic tasks, the low-frequency (0-10%) energy distribution remains relatively uniform with little variation among the three tasks.

**Performance under Frequency Compression.** To further assess the role of high-frequency information on task performance across different benchmarks, we compress action signals by removing the high-frequency components from the original sequences, reconstructed the actions with varying proportions of low-frequency information, and measured their success rates on the original tasks. The results, shown in Figure 3*(b)*, indicate that omitting high-frequency signals leads to a significant drop in success rates for Adroit tasks. In contrast, low-dimensional tasks in Robomimic maintain stable performance with as little as 12% of low-frequency information preserved. This analysis underscores the importance of frequency-aware modeling: the necessity of high-frequency signals for effective action representation varies across different scenarios.

**Conclusion.** These findings provide the empirical foundation for our frequency-based autoregressive generation strategy. They suggest that adjusting the granularity of frequency-domain representations can help optimize model performance for tasks with different levels of complexity. Our goal is to enable the model to learn individual task skills from expert demonstrations in diverse action spaces and with varying task complexities. Therefore, we first transform the actions into the frequency domain and recover them at different frequency scales. This allows the model to predict subsequent actions using information from lower frequency bands, enabling a coarse-to-fine generation process.

### 3.3 DCT Decomposition

Our spectral analysis reveals that the proportion of frequency components required for effective action representation varies across different tasks. Therefore, it is necessary to transform the action space from the time domain to the frequency domain, for more effective robotic action modeling. To efficiently achieve this, we employ the Discrete Cosine Transform (DCT), which can project

time-domain trajectories onto a set of cosine basis functions with low computational cost. This provides a compact and interpretable frequency-domain representation, enabling hierarchical modeling of actions in various tasks settings and scenarios. Specifically, given an action sequence $\mathbf{x} = [x_0, x_1, \ldots, x_{N-1}]^T \in \mathbb{R}^{N \times d}$, where each $x_n \in \mathbb{R}^d$ represents the action at time step $n$, we transform it into the frequency domain by applying the DCT independently to each action dimension. For each dimension, the DCT is defined as:

$$X_i = \sum_{n=0}^{N-1} x_n \cos \left[ \frac{\pi}{N} \left( n + \frac{1}{2} \right) i \right], \quad i = 0, 1, \ldots, N-1 \tag{1}$$

Here, $X_i$ denotes the $i$-th DCT coefficient. Given any integer $k$ such that $1 \leq k \leq N$, we refer to retaining only the first $k$ DCT coefficients as preserving the $k$-level DCT. With these $k$ coefficients, the original trajectory can be approximately reconstructed using the inverse DCT, as shown in Equation 2.

$$y_n^k = \frac{1}{N} \left[ X_0 + 2 \sum_{i=1}^{k-1} X_i \cos \left( \frac{\pi}{N} \left( n + \frac{1}{2} \right) i \right) \right], \quad n = 0, 1, \ldots, N-1 \tag{2}$$

Specifically, let $\mathbf{y}^k = [y_0^k, y_1^k, \ldots, y_{N-1}^k]^T \in \mathbb{R}^{N \times d}$ denote the $k$-level reconstruction of the original sequence $\mathbf{x}$. If $k = N$, meaning all DCT coefficients are used, the reconstruction will recover the original trajectory without any loss of information. However, if $k < N$, only the lowest $k$ frequency components are kept, and all higher-frequency components are discarded. As a result, the reconstructed trajectory $\mathbf{y}^k$ becomes a compressed and smoother version of the original, preserving the essential structure of the original sequence while discarding minor fluctuations.

### 3.4 Continuous Tokens for FreqPolicy

We explore integrating FreqPolicy with continuous token representations using a diffusion model. Compared to discrete tokens, continuous tokens offer greater expressiveness and enable more fine-grained modeling of the action space, allowing for smoother and more accurate representation of complex trajectories. Accurately modeling the probability distribution of each token is essential for effectively incorporating a continuous tokenizer into autoregressive models. Building on MAR [19], we tackle this challenge by adopting a diffusion-based loss function for training and devising a specialized frequency-aware sampling mechanism for efficient inference. Additional details on our training and inference processes are provided in the appendix.

**Training** During training, given an action sequence $\mathbf{x} \in \mathbb{R}^N$ of length $N$, we first apply the Discrete Cosine Transform (DCT) to obtain its frequency coefficients, as defined in Equation 1, and compute reconstructions at all frequency levels as described in Equation 2. Our objective is to train a model capable of modeling token distributions across different frequency levels. For an arbitrary frequency level $k$, we encode the $k$-level reconstruction $\mathbf{y}^k$, the observation $\mathbf{o}$, and the level index $k$ using the FreqPolicy encoder to obtain a latent representation. This latent feature is then decoded by the FreqPolicy decoder to produce a condition vector $\mathbf{z}^k$, which serves as a continuous token for $\mathbf{y}^k$. We subsequently model the conditional probability $p(\mathbf{x} \mid \mathbf{z}^k)$ using a reverse diffusion process, which allows for conditional generation based on $\mathbf{z}^k$ to recover the original action sequence. This recovered sequence can then be further processed with DCT and inverse DCT to obtain reconstructions at any desired frequency level.

**Sampling** The sampling procedure follows the standard inference process of diffusion models. Beginning with initial noise sampled from a standard normal distribution, the diffusion tokenizer iteratively removes the noise to produce $\hat{\mathbf{x}} \sim p(\mathbf{x} \mid \mathbf{z}^k)$. Here, $\hat{\mathbf{x}}$ denotes a trajectory sampled from the action space of demonstration data with complete frequency components. This enables flexible reconstructions at various frequency levels, by simply specifying the desired frequency level $k$.

**Diffusion Loss** The diffusion tokenizer is optimized using the diffusion loss introduced by [19], as presented in Equation 4.

$$L(z^k, \mathbf{x}) = \mathbb{E}_{\epsilon, t} \left[ \left| \epsilon - \varepsilon_\theta(\mathbf{x}_t \mid t, z^k) \right|^2 \right], \tag{3}$$

where $\mathbf{x}_t$ is a noise-perturbed version of the original trajectory $\mathbf{x}$, $\epsilon$ is a noise vector sampled from $\mathcal{N}(\mathbf{0}, \mathbf{I})$, and $\varepsilon_\theta$ is a small MLP parameterized by $\theta$ that predicts the added noise. Gradients are propagated through $z^k$, enabling end-to-end optimization of FreqPolicy encoder, decoder and the noise predictor.

**Masked Generative Strategy** To reduce training cost and enhance generation diversity, we adopt

the frequency-aware masking mechanism from FAR [45]. Since early autoregressive steps process information-sparse, low-frequency inputs, using all tokens is redundant. The frequency-aware masking strategy in FAR [45] applies higher mask ratios at lower-frequency levels, gradually increasing the number of tokens as higher-frequency information is incorporated.

### 3.5 Frequency-based Hierarchical Generation

Our previous analysis demonstrated that compressing actions in the frequency domain at different scales reveals hierarchical information. Actions reconstructed with higher compression ratio are smoother and better capture the overall trends of the motion, which helps reduce noise interference and lowers the difficulty of generation. Based on this observation, we propose a frequency-based autoregressive strategy for hierarchical action generation, as illustrated in Figure 2. We define an increasing frequency level sequence $\{l_0, l_1, \ldots, l_{N_{\text{iter}}}\}$ where $l_0 = 0$ (starting from zero input) and $l_{N_{\text{iter}}} = N$ (full frequency). Each $l_i$ represents the number of DCT coefficients retained at iteration $i$. Starting from $l_0$, we iterate from low to high frequencies $N_{\text{iter}}$ times to generate the action sequence. In each iteration $i \in \{1, 2, ..., N_{\text{iter}}\}$, the observation tokens, the frequency level index $l_{i-1}$, and the corresponding $l_{i-1}$-level reconstruction action tokens $\mathbf{y}^{l_{i-1}}$ are encoded to produce a continuous token $\mathbf{z}^{l_{i-1}}$ that represents the current context. The diffusion model then samples a full-frequency action sequence $\hat{\mathbf{x}}^i$ conditioned on this token. For the next step, we use the $l_i$-level reconstruction of $\hat{\mathbf{x}}^i$, i.e., $\mathbf{y}^{l_i}$, as the input for the following iteration, which contains richer frequency information than the previous input.

**Example.** For a 16-timestep trajectory with $N_{\text{iter}} = 4$ iterations, we might use $\{l_0 = 0, l_1 = 4, l_2 = 8, l_3 = 12, l_4 = 16\}$. At iteration 1, we start with $\mathbf{y}^0$ (all zeros) and generate $\hat{\mathbf{x}}^1$, then filter it to keep only the first 4 DCT coefficients, obtaining $\mathbf{y}^4$ for iteration 2. This process continues, progressively adding more frequency components ($4 \rightarrow 8 \rightarrow 12 \rightarrow 16$) until the final full-frequency output.

This hierarchical generation paradigm first captures the basic trends and global structure of the action sequence using low-frequency components, and then progressively enriches the output by incorporating higher-frequency signals that model fine-grained motion details. By explicitly decomposing and modeling motions at different frequency levels, our approach allows the model to better learn and represent both global dynamics and subtle variations in complex action sequences. This structured frequency decomposition, combined with autoregressive conditioning, enables more efficient and expressive generation of diverse and realistic motions.

## 4 Experiments

This section provides a comprehensive evaluation of our proposed method. We first describe the experimental setup, including benchmarks, baseline methods, and implementation details. Next, we analyze the frequency domain requirements of different tasks and highlight their unique characteristics. We then compare our method in both the time and frequency domains. Additionally, we benchmark our approach against autoregressive methods with continuous and discrete token representations across various simulation benchmarks, and present results from real-world applications. Finally, we discuss the inference speed and ablation in sampling.

### 4.1 Experimental Setup

**Benchmarks.** We evaluate our methods on a diverse set of benchmarks that provide different types of observation data. Benchmarks with only 2D image observations are referred to as **2D tasks**, which include two single-task benchmarks, **Robomimic** [23] and **Push T** [12]. Benchmarks with 3D visual observations are referred to as **3D tasks**, consisting of **Adroit** [30], **DexArt** [4], **MetaWorld** [46], and **RobTwin** [26], which together cover a wide range of robotic manipulation and dual-arm collaborative tasks. Tasks in DexArt are conducted using Allegro Hand [3] with 22 DoF, tasks in Adroit use ShadowHand [31] with 26-28 DoF, and tasks in RoboTwin using dual-arm grippers [2] with 14 DoF. Other tasks use parallel grippers with 10 DoF.

**Baseline.** For **2D tasks**, we compare against Diffusion Policy (DP) as well as two autoregressive approaches using discrete token representation: Behavior Transformer (BeT) [32] and CARP [16]. DP is available in two variants: CNN-based (DP-C) and Transformer-based (DP-T). For **3D tasks**, we use 3D Diffusion Policy (DP3) [48] and Mamba Policy [9] as baselines.

**Implementation Details.** Our model can be seamlessly integrated into the codebases of Diffusion Policy (DP) and 3D Diffusion Policy (DP3). To ensure fair comparisons, we use the same parameters and observation input processing as Diffusion Policy for the **2D tasks**, and as 3D Diffusion Policy for the **3D tasks**, maintaining consistency with their respective frameworks. Our approach enables flexible adjustment of autoregressive iteration counts in the frequency domain during inference. For all simulation experiments, we use 4 iterations, whereas for real-world experiments, only 1 iteration is used.

Table 1: **Success rate (%) comparison on 10 simulation tasks in Adroit, DexArt and Meta-World.** To ensure fair comparison, all tasks employ full-spectrum outputs and identical frequency level progression. * denotes results we reproduced using the same expert demonstrations as ours.

| Alg \ Task | Adroit [30] | | | DexArt [4] | | | | Meta-World [46] | | | Average |
|---|---|---|---|---|---|---|---|---|---|---|---|
| | Hammer | Door | Pen | Laptop | Faucet | Toilet | Bucket | Assembly | Disassemble | Stick Push | |
| Diffusion Policy | 45±5 | 37±2 | 13±2 | 69±4 | 23±8 | 58±2 | 46±1 | 15±1 | 43±7 | 63±3 | 44.0 |
| DP3 | 100±0 | 62±4 | 43±6 | 83±1 | 63±2 | 82±4 | 46±2 | 99±1 | 69±4 | 97±4 | 74.4 |
| Mamba Policy | 100±0 | 68±1 | 41±2 | 80±4 | 33±2 | 76±0 | 27±1 | 100±0 | 76±4 | 100±0 | 70.1 |
| DP3* | **100±0** | 53±2 | 50±5 | 83±3 | 33±2 | 70±6 | 24±4 | 95±3 | 87±3 | 83±3 | 67.8 |
| Mamba Policy* | **100±0** | 59±3 | 55±2 | 79±3 | **35±6** | 65±5 | 23±2 | 96±2 | 90±2 | 82±5 | 68.4 |
| ours(w/o DCT) | **100±0** | 51±5 | 47±3 | 55±6 | 23±2 | 45±5 | **30±3** | 95±0 | 88±3 | 80±3 | 61.4 |
| ours | **100±0** | **65±5** | **59±5** | **85±4** | 30±3 | **77±3** | 25±3 | **97±2** | **92±6** | **85±5** | **71.5** |

Table 2: **Main results on 48 simulation tasks.** The table reports average success rates (%) across all tasks. * denotes results we reproduced using the same expert demonstrations for fair comparison. Success rates for individual tasks are in Appendix.

| Algorithm \ Task | Adroit (3) | DexArt (4) | MetaWorld Easy (20) | MetaWorld Medium (11) | MetaWorld Hard (5) | MetaWorld Very Hard (5) | Average |
|---|---|---|---|---|---|---|---|
| DP3 | 68.3 | 68.5 | 91.7 | 61.6 | 38.0 | 49.0 | 62.9±16.9 |
| Diffusion Policy | 31.7 | 49.0 | 86.8 | 31.1 | 10.8 | 26.6 | 39.3±23.9 |
| DP3* | 67.7 | 52.5 | 89.9 | 66.8 | 42.8 | 68.0 | 64.6±14.7 |
| Mamba Policy* | 71.3 | 50.5 | 90.2 | **67.4** | 46.4 | 69.0 | 65.8±14.4 |
| ours | **74.7** | **54.3** | **92.4** | **67.4** | **48.8** | **70.2** | **67.9±14.2** |

Table 3: **Success rate (%) comparison on the RoboTwin Benchmark for Dual-Arm Manipulation with D435 Camera Setting.** We evaluated our approach on 7 tasks using 20 expert demonstrations and 3 seeds (0, 1, 2) and report the success rate. Our method was compared against DP3 (XYZ+RGB) and DP, all tested under the same conditions. Both DP and our method are trained for 500 epochs, while DP3 is trained for 3,000 epochs.

| Task | RoboTwin [26] | | | | | | |
|---|---|---|---|---|---|---|---|
| | Block Hammer Beat | Block Handover | Bottle Adjust | Container Place | Empty Cup Place | Pick Apple Messy | Dual Bottles Pick(Hard) |
| DP | 0.0±0.0 | 0.0±0.0 | 6.3±5.9 | 1.7±0.6 | 0.0±0.0 | 5.3±2.5 | 8.0±2.0 |
| DP3(XYZ+RGB) | **47.7±4.0** | **86.0±1.0** | 25.0±5.0 | 37.3±2.1 | 23.7±5.5 | 6.0±2.6 | 28.0±4.4 |
| Ours(XYZ+RGB) | 42.0±4.2 | 80.7±9.7 | **27.7±11.4** | **39.7±3.3** | **29.3±9.4** | **7.0±1.0** | **30.0±4.2** |

## 4.2 Simulation Result

**High Success Rate.** In Table 1, we compare our method on 10 tasks from Adroit, DexArt, and MetaWorld, while Table 3 presents results on 7 RoboTwin tasks using colored point clouds as observations. In Table 1, our method outperforms both the state-of-the-art (SOTA) methods and a baseline (without DCT decomposition) on 8 tasks, and achieves comparable results on the remaining 2. Similarly, as shown in Table 3, our method achieves superior performance in RoboTwin, with 5 tasks reaching higher success rates than DP and DP3. We also evaluated all 48 tasks in Adroit, DexArt, and MetaWorld and reported the average success rate for each benchmark. As shown in Table 2, our approach maintains an average success rate of 67.9%, surpassing Diffusion Policy 3D (64.6%) and Mamba Policy (65.8%). These results demonstrate the effectiveness of our approach across a variety of task settings and scenarios. *More detailed results for each individual task are provided in the appendix.*

**Generalization Ability.** Results in Table 4 demonstrate that our method achieves consistently strong performance in unseen DexArt test environments, maintaining high success rates under both limited (10 demonstrations) and abundant (100 demonstrations) data settings. These results demonstrate the method's robustness and generalization ability in diverse scenarios.

Table 4: **Generalization Results.** Success rates (%) on unseen DexArt test data after 3000 training epochs, averaged over 100 trials × 3 random seeds.

| Alg \ Task | 10 Demonstrations | | | | 100 Demonstrations | | | |
|---|---|---|---|---|---|---|---|---|
| | Laptop | Faucet | Toilet | Bucket | Laptop | Faucet | Toilet | Bucket |
| DP3 | $12.2_{\pm2}$ | $11.5_{\pm3}$ | $8.5_{\pm1}$ | $15.0_{\pm3}$ | $36.4_{\pm5}$ | $\mathbf{17.4}_{\pm2}$ | $25.4_{\pm1}$ | $\mathbf{29.2}_{\pm6}$ |
| Mamba Policy | $11.9_{\pm3}$ | $8.7_{\pm3}$ | $11.2_{\pm4}$ | $23.6_{\pm4}$ | $30.9_{\pm5}$ | $16.6_{\pm4}$ | $26.8_{\pm5}$ | $25.9_{\pm3}$ |
| ours | $\mathbf{14.5}_{\pm3}$ | $\mathbf{12.4}_{\pm4}$ | $\mathbf{16.6}_{\pm4}$ | $\mathbf{32.3}_{\pm4}$ | $\mathbf{40}_{\pm3}$ | $12.1_{\pm3}$ | $\mathbf{27.8}_{\pm3}$ | $26.4_{\pm4}$ |

Table 5: **Noise Robustness Evaluation.** Success rates (%) with relative change vs. high-quality baseline (see Appendix Table 10) shown as superscripts. We evaluate two noise conditions: (1) Low-Quality demonstrations using suboptimal trajectories (5 tasks, hammer excluded), and (2) Gaussian Noise (std=0.025, 0.05, 0.1) applied to high-quality baseline demonstrations (6 tasks).

| Method | Low-Quality Demonstrations | | | | | | | Gaussian Noise std=0.025 | | | | | | |
|---|---|---|---|---|---|---|---|---|---|---|---|---|---|---|
| | Pick | Soccer | Stick | Hammer | Door | Pen | Avg. | Pick | Soccer | Stick | Hammer | Door | Pen | Avg. |
| DP3 | $10^{-17}$ | $15^{-32}$ | $41^{-28}$ | – | $47^{-11}$ | $33^{-34}$ | $29^{-40}$ | $26^{+117}$ | $25^{+14}$ | $63^{+11}$ | $0^{-100}$ | $\mathbf{55}^{+4}$ | $53^{+6}$ | $37^{-24}$ |
| Mamba | $16^{-36}$ | $\mathbf{20}^{-29}$ | $40^{-27}$ | – | $44^{-25}$ | $\mathbf{34}^{-38}$ | $31^{-43}$ | $18^{-28}$ | $33^{+18}$ | $61^{+11}$ | $0^{-100}$ | $50^{-15}$ | $\mathbf{57}^{+4}$ | $37^{-32}$ |
| Ours | $\mathbf{20}^{-33}$ | $17^{-47}$ | $\mathbf{43}^{-31}$ | – | $\mathbf{49}^{-25}$ | $32^{-45}$ | $\mathbf{32}^{-44}$ | $\mathbf{32}^{+7}$ | $\mathbf{33}^{+3}$ | $60^{-3}$ | $\mathbf{70}^{-30}$ | $52^{-20}$ | $56^{-3}$ | $\mathbf{51}^{-13}$ |

| Method | Gaussian Noise std=0.05 | | | | | | | Gaussian Noise std=0.1 | | | | | | |
|---|---|---|---|---|---|---|---|---|---|---|---|---|---|---|
| | Pick | Soccer | Stick | Hammer | Door | Pen | Avg. | Pick | Soccer | Stick | Hammer | Door | Pen | Avg. |
| DP3 | $21^{+75}$ | $19^{-14}$ | $51^{-11}$ | $0^{-100}$ | $27^{-49}$ | $55^{+10}$ | $29^{-41}$ | $7^{-42}$ | $18^{-18}$ | $50^{-12}$ | $0^{-100}$ | $24^{-55}$ | $38^{-24}$ | $23^{-53}$ |
| Mamba | $23^{-8}$ | $25^{-11}$ | $\mathbf{60}^{+9}$ | $0^{-100}$ | $\mathbf{51}^{-14}$ | $\mathbf{57}^{+4}$ | $36^{-33}$ | $3^{-88}$ | $14^{-50}$ | $39^{-29}$ | $0^{-100}$ | $21^{-64}$ | $39^{-29}$ | $19^{-64}$ |
| Ours | $\mathbf{25}^{-17}$ | $\mathbf{30}^{-6}$ | $60^{-3}$ | $\mathbf{51}^{-49}$ | $46^{-29}$ | $55^{-5}$ | $\mathbf{45}^{-23}$ | $\mathbf{20}^{-33}$ | $\mathbf{27}^{-16}$ | $\mathbf{52}^{-16}$ | $\mathbf{15}^{-85}$ | $\mathbf{32}^{-51}$ | $\mathbf{49}^{-16}$ | $\mathbf{33}^{-44}$ |

**Noise Robustness.** We evaluate robustness under two noise scenarios (Table 5): (1) training on low-quality demonstrations with reduced reward thresholds (5 tasks used, hammer task cannot generate low-reward expert demonstrations), and (2) adding Gaussian noise (std=0.025, 0.05, 0.1) to high-quality demonstrations. On low-quality demonstrations, all methods show significant performance decline, validating the importance of high-quality training data. More importantly, under Gaussian noise conditions, our method demonstrates exceptional robustness: at high noise (std=0.1), we achieve 33% success versus 19-23% for baselines. The Hammer task is particularly revealing—baselines completely fail (0% success) under any Gaussian noise, while we maintain 70%, 51%, and 15% at increasing noise levels. This superior performance stems from our low-frequency to high-frequency progressive generation mechanism's natural anti-noise advantages: (1) *Low-frequency anti-noise characteristics*—low-frequency components inherently have stronger noise resistance, establishing a stable action foundation in initial stages; (2) *Progressive filtering*—each frequency domain level's reconstruction process acts as natural noise filtering; (3) *Structured constraints*—the coarse-to-fine generation process ensures that key action structures are preserved even under noise interference. Interestingly, moderate noise (std=0.025) can promote generalization in certain tasks: for Pen spinning and Pick Out of Hole, low-intensity Gaussian noise unintentionally increases hand movement variation, adding action diversity and leading to higher success rates than noise-free baselines. This suggests beneficial noise augmentation can enrich the action distribution.

Table 6: **Success rates (%) comparisons on Robomimic benchmark [24] and Push-T task [12].**

| Tokenizer | Model | Lift(mh) | | Can(mh) | | Square(mh) | | Transport(mh) | | Push-T | | Time |
|---|---|---|---|---|---|---|---|---|---|---|---|---|
| | | Visual | State | Visual | State | Visual | State | Visual | State | Visual | State | |
| **Diffusion Models** | | | | | | | | | | | | |
| Continuous | DP-C | **1.00**/**1.00** | **1.00**/0.97 | **1.00**/0.96 | **1.00**/0.96 | **0.98**/**0.84** | **0.97**/**0.82** | **0.89**/**0.69** | **0.68**/**0.46** | **0.91**/**0.84** | **0.95**/**0.91** | 2.11 |
| | DP-T | **1.00**/0.99 | **1.00**/**1.00** | **1.00**/0.98 | **1.00**/0.94 | 0.94/0.80 | 0.95/0.81 | 0.73/0.50 | 0.62/0.35 | 0.78/0.66 | **0.95**/0.79 | 1.35 |
| **Autoregressive Models** | | | | | | | | | | | | |
| Discrete | BET | – | **1.00**/0.99 | – | **1.00**/0.90 | – | 0.68/0.43 | – | 0.21/0.06 | – | 0.79/0.70 | **0.007** |
| | CARP | 0.94/0.90 | **1.00**/0.97 | 0.74/0.68 | 0.88/0.85 | 0.46/0.42 | 0.44/0.37 | 0.00/0.00 | 0.00/0.00 | 0.88/0.83 | 0.85/0.83 | 0.09 |
| Continuous | Ours | **1.00**/**1.00** | **1.00**/0.98 | 0.98/0.94 | **1.00**/0.90 | 0.84/0.78 | 0.88/0.74 | 0.58/0.50 | 0.50/0.38 | 0.82/0.76 | 0.92/0.85 | 0.21 |

**Discrete vs Continuous.** As shown in Table 6, our systematic comparison between discrete autoregressive and continuous diffusion models reveals a key finding: discrete representation methods (CARP, BET) exhibit significant limitations in modeling continuous action spaces, primarily due to information loss during the discretization process, particularly evident in hard tasks requiring precise control. In contrast, our proposed approach that integrates autoregressive and diffusion mechanisms in a continuous representation framework not only maintains consistent high performance across various tasks, but also inherits the computational efficiency advantages of autoregressive methods (requiring only 1/10 of DP's inference time), conclusively demonstrating its superior expressiveness,

computational efficiency, and environmental adaptability in modeling continuous action spaces.

**Inference Efficiency.** As shown in Table 6, our continuous token AR method with integrated diffusion mechanisms achieves an order of magnitude faster inference (0.21s vs. 2.11s) compared to diffusion policy while maintaining comparable success rates on the Robomimic benchmark.

**Flexible Sampling.** Our method allows flexible selection of the number of autoregressive iterations in the frequency domain during inference. We evaluated the impact of different iteration counts on task success rates and inference times on the Adroit benchmark, and provide a Pareto frontier visualization [22] in Figure 4, which displays the tradeoff between inference time and success rate, methods closer to the top-left corner achieve better trade-offs between efficiency and performance. It shows that our method maintaining competitive performance even at minimal iterations.

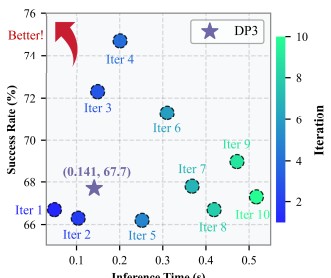

Figure 4: **Pareto Analysis on Adroit benchmark.** The x-axis represents inference time and the y-axis indicates task success rate.

### 4.3 Real-World Result

To validate our method in real-world applications, we collected demonstration data for an object handover task using the teleoperation system of the ShadowHand [31]. In this setup, a human subject holds an object, and the robot is required to stably receive it. The dynamic interactions between the hand and object necessitate high inference speed for real-time responsiveness. Our real-world tests, conducted on an RTX 4090 GPU, show that our single-iteration implementation achieves 70 FPS, significantly outperforming DP3's 25 FPS. As illustrated in Figure 5, our robotic hand successfully receives the object from the human subject. These results demonstrate the practical advantages of our method for real-time robotic control applications with strict temporal constraints. *Additional experimental details for real-world scenarios are provided in the appendix.*

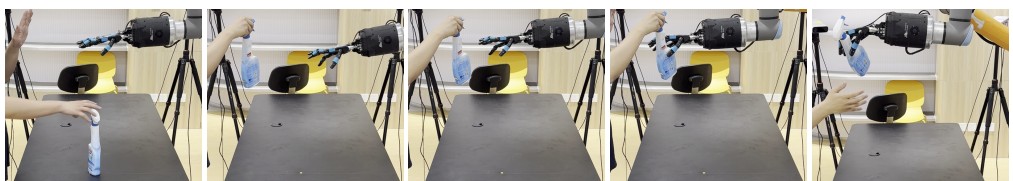

Figure 5: **Real-World Experiments on Robotic Handover Task.** The robotic hand stably receives an object from a human subject during real-world testing.

## 5 Conclusion

In this paper, we introduced FreqPolicy, a novel visuomotor policy framework leveraging hierarchical frequency-domain representations and continuous tokens for effective robotic manipulation. By decomposing action signals into frequency components using DCT, our method allows for flexible reconstruction of actions under different levels of action details. By integrating continuous latent representations with an autoregressive paradigm, our method enables precise and efficient modeling of action spaces via coarse-to-fine generation, eliminating discretization losses. Extensive experiments demonstrate that our approach achieves state-of-the-art performance, outperforming existing methods in both success rate and computational efficiency. In future work, we plan to extend our research to the VLA framework, further exploring how frequency domain representations influence action spaces in multi-task training environments.

## 6 Limitations

Since all of our experiments used condition inputs consistent with DP3 or DP, we have not yet explored the potential impact of altering condition input methods on model performance. Additionally, it should be noted that our method still has room for improvement in 2D tasks, and performance tends to decrease when frequency domain partitioning becomes too fine-grained.

# 7  acknowledgements

This work was supported by NSFC (No.62206173), Shanghai Frontiers Science Center of Human-centered Artificial Intelligence (ShangHAI), and MoE Key Laboratory of Intelligent Perception and Human-Machine Collaboration (KLIP-HuMaCo).

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

# Appendix

This document contains supplementary materials for our main paper. We provide further technical details, additional experimental results, and more qualitative examples to complement the findings presented in the main text. We hope this supplementary information will help readers better understand our approach and results.

The remainder of this supplementary material is organized as follows. In Section A, we provide the hardware specifications used in our experiments. In Section B, we list the hyperparameters employed. Section C presents the detailed algorithms for training and inference. In Section D, we describe the ablation studies conducted. Section E outlines the details of our real-world experiments. Finally, in Section F, we offer further discussion on VLA models and frequency domain analysis.

## A    Computational Resources

To ensure reproducibility, we provide detailed information on the computational resources used in our experiments. For all simulation environment experiments including training, inference, and time benchmarking tests, we used NVIDIA RTX 2080Ti GPUs. Our model has 63M parameters, with DP3 at 255M, consuming approximately 4.5GB of memory during operation.For real-world environment experiments, we employed NVIDIA RTX 4090 GPUs for training, inference, and time benchmarking tests.

## B    Hyperparameters

In Table 7, we present the hyperparameters used in our experiments. For the baseline methods DP and DP3, we use their default hyperparameters. For the Adroit, DexArt, and MetaWorld benchmarks, our models are trained for 3,000 epochs. For the Robomimic and Push-T tasks, we use 1,000 training epochs, and for the RoboTwin benchmark, our models are trained for 500 epochs.

Table 7: **Hyperparameters used for various benchmark.**

| Hyperparameter | Value |
|---|---|
| Horizon ($T_h$) | 16 (8 for RoboTwin) |
| Action step ($T_a$) | 8 (6 for RoboTwin) |
| Observation step ($T_o$) | 2 (3 for RoboTwin) |
| point_feature_dim | 64 |
| state_mlp_size | 64 |
| Batchsize | 128 |
| Num_iter | 4 |
| Num_training_steps(Diffusion training) | 100 |
| Num_sampling_steps(Diffusion sampling) | ddim10 |
| Diffloss_d | 3 |
| Diffloss_w | 1024 |
| encoder_embed_dim | 512 |
| decoder_embed_dim | 512 |
| encoder_depth | 4 |
| decoder_depth | 4 |
| encoder_num_heads | 8 |
| decoder_num_heads | 8 |
| Optimizer | AdamW |
| Betas ($\beta_1, \beta_2$) | [0.95, 0.999] |
| Learning Rate | 1.0e-4 |
| Weight Decay | 1.0e-6 |
| Learning Rate Scheduler | Cosine |

## C   Training and Inference Details

The training process for our FreqPolicy is outlined in Algorithm 1. At each epoch, we first encode the input observations using an observation encoder. The ground truth action sequence is then transformed into the frequency domain via the Discrete Cosine Transform (DCT), and a frequency index is randomly sampled. Conditional reconstruction is performed by applying the inverse DCT up to the sampled frequency level, enabling the model to focus on different frequency components during training. An adaptive mask ratio is determined based on the frequency index, and a mask is sampled accordingly. The masked observation and conditional reconstruction are then encoded and subsequently decoded by the FreqPolicy encoder and decoder, respectively. The diffusion model is trained to predict noise added to the actions at randomly sampled diffusion steps, using a standard mean squared error loss. Model parameters are updated by minimizing this loss throughout the training epochs. This procedure effectively leverages masked autoregressive modeling and diffusion-based generation, enabling the policy to learn robust representations across the frequency domain.

---

**Algorithm 1** FreqPolicy Training

---

**Require:** Number of training epochs $K$, Observation Encoder $\mathcal{E}_{obs}$, FreqPolicy Encoder $\mathcal{E}$, FreqPolicy Decoder $\mathcal{D}$, diffusion model $\epsilon_\theta$, Observation $\mathcal{O}$, Ground truth action $\mathbf{x}$, Horizon $T$, diffusion steps $T_{diff}$, initial mask ratio $m$.

1: **for** $e = 1$ to $K$ **do**
2:    $z_{obs} \leftarrow \mathcal{E}_{obs}(\mathcal{O})$                    ▷ Encode observations
3:    $\{X_0, X_1, \ldots, X_{T-1}\} \leftarrow \text{DCT}(\mathbf{x}),\ k \sim \mathcal{U}(0, T)$      ▷ Apply DCT, sample index
4:    $y^k \leftarrow \begin{cases} \text{IDCT}(\{X_0, X_1, \ldots, X_{k-1}\}) & \text{if } k > 0 \\ \mathbf{0} & \text{if } k = 0 \end{cases}$      ▷ Extract k-level reconstruction
5:    $\text{mask\_ratio} \leftarrow m \cdot (1 - k/T)$                ▷ Adaptive mask ratio
6:    $\text{mask} \sim \text{TruncNorm}(\text{mask\_ratio})$
7:    $z_{mask} \leftarrow \mathcal{E}(z_{obs}, y^k, k, \text{mask})$            ▷ Encode with mask
8:    $z^k \leftarrow \mathcal{D}(z_{obs}, z_{mask}, k, \text{mask})$            ▷ Decode with mask
9:    $t \sim \mathcal{U}(1, T_{diff}),\ \epsilon \sim \mathcal{N}(0, \mathbf{I})$
10:   $\mathbf{x}_t \leftarrow \sqrt{\bar{\alpha}_t}\mathbf{x} + \sqrt{1 - \bar{\alpha}_t}\epsilon$
11:   $\mathcal{L} \leftarrow \mathbb{E}_{\epsilon,t}[\|\epsilon - \epsilon_\theta(\mathbf{x}_t, t, k, z^k)\|^2]$            ▷ Diffusion loss
12:   Update model parameters by minimizing $\mathcal{L}$
13: **end for**
14: **return** Trained models $\mathcal{E}_{obs}$, $\mathcal{E}$, $\mathcal{D}$, and $\epsilon_\theta$

---

The inference process for FreqPolicy is detailed in Algorithm 2. Given an input observation, we first encode it using the observation encoder. The action tokens are initialized to zeros and are fully masked at the beginning. For each iteration, the model progressively predicts actions at increasing frequency levels, guided by the current frequency index.

At each step $k$, the partially reconstructed tokens and the current mask are passed through the encoder and decoder to obtain the continuous latent code $z^k$. The diffusion sampler then generates an updated action prediction conditioned on $z^k$. If it is not the final step, the predicted actions are transformed into the frequency domain using the Discrete Cosine Transform (DCT), and the reconstruction is refined up to the next frequency level via inverse DCT. The masking ratio is adaptively reduced at each iteration, gradually revealing more of the reconstructed action sequence. This process continues until the entire action sequence is fully reconstructed. By progressively incorporating information across different frequency bands, our inference procedure enables the policy to generate high-fidelity predictions.

## D   Ablation

To verify the effectiveness of our proposed method and the contribution of each component, we conducted a series of ablation experiments.

The results in Table 8 present the ablation study on the prediction horizon, it shows that the performance is not sensitive to this parameter, demonstrating the robustness of our method. When $T_h = 8$

**Algorithm 2** FreqPolicy Inference

**Require:** Observation Encoder $\mathcal{E}_{obs}$, FreqPolicy Encoder $\mathcal{E}$, FreqPolicy Decoder $\mathcal{D}$, DiffusionSampler $\mathcal{F}$, diffusion steps $T_{diff}$, Observation $\mathcal{O}$, Horizon $T$, Number of iterations $N_{iter}$, Frequency indices $\{i_0, i_1, \ldots, i_{N_{iter}-1}\}$.

1: **Initialize:**
2:     mask $\leftarrow \mathbf{1}_{B \times T}$        ▷ Full masking initially
3:     tokens $\leftarrow \mathbf{0}_{B \times T \times D_{action}}$        ▷ Zero initialization
4:     $z_{obs} \leftarrow \mathcal{E}_{obs}(\mathcal{O})$        ▷ Encode observations
5: **for** step $= 0$ to $N_{iter} - 1$ **do**
6:     $k \leftarrow i_{\text{step}}$        ▷ Current frequency index
7:     $z_{mask} \leftarrow \mathcal{E}(z_{obs}, \text{tokens}, k, \text{mask})$
8:     $z^k \leftarrow \mathcal{D}(z_{obs}, z_{mask}, k, \text{mask})$
9:     $\hat{\mathbf{x}} \leftarrow \mathcal{F}(z^k, k, T_{diff})$        ▷ Generate prediction via diffusion
10:     **if** step $< N_{iter} - 1$ **then**
11:        $\{X_0, X_1, \ldots, X_{T-1}\} \leftarrow \text{DCT}(\hat{\mathbf{x}})$        ▷ Transform to frequency domain
12:        next_k $\leftarrow i_{\text{step}+1}$        ▷ Next frequency level
13:        tokens $\leftarrow \text{IDCT}(\{X_0, X_1, \ldots, X_{\text{next\_k}-1}\})$
14:     **else**
15:        tokens $\leftarrow \hat{\mathbf{x}}$
16:     **end if**
17:     mask_ratio $\leftarrow \cos\left(\frac{\pi}{2} \cdot \frac{\text{step}+1}{N_{iter}}\right)$
18:     mask $\leftarrow \text{GenerateMask}(\text{mask\_ratio})$
19: **end for**
20: **return** tokens        ▷ Final action sequence

or $T_h = 16$, the model achieves slightly better performance with an average score over 58 points, while both smaller and larger horizons yield comparable results. This robustness across a range of temporal scales highlights the flexibility of our approach. Notably, only at extremely long horizons ($T_h = 64$) do we observe a noticeable decline in performance.

Table 9 presents the results of the ablation study on masking strategies. Our frequency policy mask significantly improves performance across all tasks, increasing the average score from 40 to 58—an improvement of approximately 45%. These results clearly demonstrate the importance of the frequency-based masking strategy for prediction tasks.

Table 8: **Ablation study on prediction horizon.** Analysis of Horizon ($T_h$), Action step ($T_a$), and Observation step ($T_o$).

| $T_h$ | $T_o$ | $T_a$ | Hammer | Door | Pen | Pick Out of Hole | Soccer | Stick Pull | **Average** |
|---|---|---|---|---|---|---|---|---|---|
| 4 | 2 | 1 | 65±6 | **76±3** | 55±5 | **37±4** | 23±4 | 60±2 | 53±17 |
| 4 | 2 | 2 | 62±4 | 71±4 | 53±6 | 31±3 | **38±3** | **64±2** | 53±14 |
| 8 | 2 | 4 | **100±0** | 68±2 | 52±4 | 25±2 | **38±4** | 62±0 | **58±24** |
| 8 | 2 | 6 | **100±0** | 72±4 | 50±5 | 29±3 | 31±2 | **64±3** | **58±25** |
| 16 | 2 | 8 | **100±0** | 65±5 | **59±5** | 30±2 | 32±4 | 62±0 | **58±23** |
| 16 | 2 | 12 | **100±0** | 59±4 | 51±3 | 35±2 | 35±4 | 55±5 | 56±21 |
| 32 | 2 | 16 | 98±2 | 58±5 | 38±2 | 34±3 | 19±2 | 52±4 | 50±25 |
| 64 | 2 | 32 | 80±4 | 35±4 | 42±2 | 35±5 | 32±3 | 38±4 | 44±17 |

Table 9: **Ablation study on mask.** This experiment analyzes the model performance with and without mask.

| | Hammer | Door | Pen | Pick Out of Hole | Soccer | Stick Pull | **Average** |
|---|---|---|---|---|---|---|---|
| W/o mask | 99±1 | 35±4 | 31±3 | 15±0 | 27±2 | 34±2 | 40±27 |
| Freqpolicy | **100±0** | **65±5** | **59±5** | **30±2** | **32±4** | **62±0** | **58±23** |

Table 10: **Main results on 48 simulation tasks.** Success rates (%) for each task are provided in this table.

| Alg \ Task | Button Press | Button Press Wall | Coffee Button | Dial Turn | Door Close | Reach Wall | Door Open | Door Unlock | Drawer Close | Drawer Open |
|---|---|---|---|---|---|---|---|---|---|---|
| | | | | | **Meta-World [46] (Easy)** | | | | | |
| DP3 | 100±0 | 99±1 | 100±0 | 66±1 | 100±0 | 68±3 | 99±1 | 100±0 | 100±0 | 100±0 |
| Diffusion Policy | 99±1 | 97±3 | 99±1 | 63±10 | 100±0 | 59±7 | 98±3 | 100±0 | 100±0 | 93±3 |
| DP3* | 100±0 | 100±0 | 100±0 | 58±5 | 100±0 | 47±5 | 100±0 | 100±0 | 100±0 | 100±0 |
| Mamba Policy* | 100±0 | 100±0 | 100±0 | 56±4 | 100±0 | 50±3 | 100±0 | 100±0 | 100±0 | 100±0 |
| ours | 100±0 | 100±0 | 100±0 | 72±4 | 100±0 | 71±4 | 100±0 | 100±0 | 100±0 | 100±0 |

| Alg \ Task | Faucet Open | Handle Press | Lever Pull | Plate Slide | Plate Slide Back | Plate Slide Back Side | Plate Slide Side | Reach | Window Close | Window Open |
|---|---|---|---|---|---|---|---|---|---|---|
| | | | | | **Meta-World (Easy)** | | | | | |
| DP3 | 100±0 | 100±0 | 79±8 | 100±1 | 99±0 | 100±0 | 100±0 | 24±1 | 100±0 | 100±0 |
| Diffusion Policy | 100±0 | 81±4 | 49±5 | 83±4 | 99±0 | 100±0 | 100±0 | 18±2 | 100±0 | 100±0 |
| DP3* | 100±0 | 86±5 | 84±2 | 100±0 | 100±0 | 100±0 | 100±0 | 22±4 | 100±0 | 100±0 |
| Mamba Policy* | 100±0 | 83±5 | 74±6 | 100±0 | 100±0 | 100±0 | 100±0 | 17±3 | 100±0 | 100±0 |
| ours | 100±0 | 90±3 | 84±4 | 100±0 | 100±0 | 100±0 | 100±0 | 30±2 | 100±0 | 100±0 |

| Alg \ Task | Hammer | Peg Insert Side | Push Wall | Soccer | Sweep | Sweep Into | Basketball | Bin Picking | Box Close | Coffee Pull | Coffee Push |
|---|---|---|---|---|---|---|---|---|---|---|---|
| | | | | | **Meta-World (Medium)** | | | | | | |
| DP3 | 76±4 | 69±7 | 49±8 | 18±3 | 96±3 | 15±5 | 98±2 | 34±30 | 42±3 | 87±3 | 94±3 |
| Diffusion Policy | 15±6 | 34±7 | 20±3 | 14±4 | 18±8 | 10±4 | 85±6 | 15±4 | 30±5 | 34±7 | 67±4 |
| DP3* | 80±5 | 62±5 | 87±5 | 22±3 | 100±0 | 17±2 | 100±0 | 35±10 | 50±6 | 95±0 | 87±5 |
| Mamba Policy* | 90±5 | 63±4 | 92±2 | 28±3 | 95±5 | 15±3 | 100±0 | 26±3 | 48±10 | 96±2 | 89±3 |
| ours | 96±1 | 51±4 | 97±3 | 32±4 | 85±4 | 19±5 | 83±8 | 31±2 | 56±4 | 100±0 | 91±4 |

| Alg \ Task | Assembly | Hand Insert | Pick Out of Hole | Pick Place | Push | Shelf Place | Disassemble | Stick Pull | Stick Push | Pick Place Wall |
|---|---|---|---|---|---|---|---|---|---|---|
| | | **Meta-World (Hard)** | | | | | **Meta-World (Very Hard)** | | | |
| DP3 | 99±1 | 14±4 | 14±9 | 12±4 | 51±3 | 17±10 | 69±4 | 27±8 | 97±4 | 35±8 |
| Diffusion Policy | 15±1 | 9±2 | 0±0 | 0±0 | 30±3 | 11±3 | 43±7 | 11±2 | 63±3 | 5±1 |
| DP3* | 95±3 | 11±2 | 12±7 | 42±6 | 54±4 | 31±4 | 87±3 | 57±3 | 83±3 | 82±6 |
| Mamba Policy* | 96±2 | 15±3 | 25±5 | 36±10 | 60±4 | 38±4 | 90±2 | 55±2 | 82±5 | 80±5 |
| ours | 97±2 | 17±3 | 30±2 | 37±4 | 63±2 | 27±3 | 92±6 | 62±0 | 85±5 | 85±3 |

| Alg \ Task | **Adroit [30]** | | | **DexArt [4]** | | | | **Average (41+3+4)** |
|---|---|---|---|---|---|---|---|---|
| | Hammer | Door | Pen | Laptop | Faucet | Toilet | Bucket | |
| DP3 | 100±0 | 62±4 | 43±6 | 83±1 | 63±2 | 82±4 | 46±2 | 71.36 |
| Diffusion Policy | 45±5 | 37±2 | 13±2 | 69±4 | 23±8 | 58±2 | 46±1 | 53.25 |
| DP3* | 100±0 | 53±2 | 50±5 | 83±3 | 33±2 | 70±6 | 24±4 | 72.91 |
| Mamba Policy* | 100±0 | 59±3 | 55±2 | 79±3 | 35±6 | 65±5 | 23±2 | 73.71 |
| ours | 100±0 | 65±5 | 59±5 | 85±4 | 30±3 | 77±3 | 25±3 | 75.54 |

# E   Real-world Experiment Details

In real-world experiments, we employ an RGB-D camera (Kinect) to capture environmental point clouds at a rate of 30 Hz. The FoundationPose model serves as our object pose estimation module, which processes the point clouds from the RGB-D camera to predict the 6D object pose, achieving a 93.22% ADD AUC score on the YCBInEOAT dataset at 30 FPS. By sampling 4,096 points from the object's mesh and applying the estimated pose from FoundationPose, we obtain cleaned object point clouds for the past 5 frames ($T_0 = 5$).

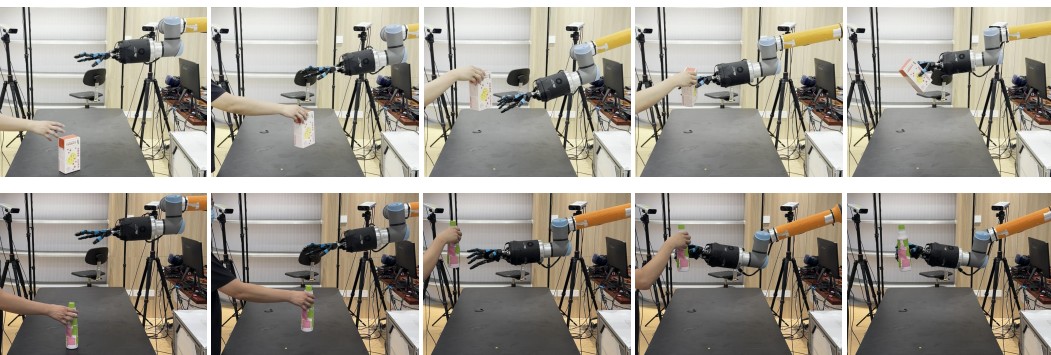

Figure 6: **Additional Visualization of Real-World Experiments on 2 Robotic Handover Tasks.**

Our algorithm takes these cleaned object point clouds along with the shadow hand poses from the previous 5 frames as input conditions, and predicts the hand's poses for the next 3 frames ($T_a = 3$). These predicted poses are executed directly on the physical robot, enabling a fully end-to-end action generation system. In terms of system performance, the perception module operates at approximately 30 FPS, while the action prediction module achieves over 70 FPS with a single iteration setting. The

complete end-to-end system maintains a comprehensive operating rate of approximately 25 FPS, meeting real-time interaction requirements. Figure 6 shows the results of two additional sets of handover tasks.

# F  Discussion

## F.1  Further Discussion on Vision-Language-Action (VLA) Models

In the main sections of this paper, we have detailed and validated the superior performance of FreqPolicy in learning specific robotic manipulation tasks. Its innovative frequency-domain autoregressive mechanism and the use of continuous tokens have demonstrated significant advantages in both precision and efficiency for single-task learning. However, a natural and promising extension is to investigate the adaptability of FreqPolicy in more complex and generalized multitask learning scenarios, especially when task instructions are given in natural language. Vision-Language-Action (VLA) models provide a powerful framework for achieving such language-driven multitask robot control.

Therefore, this section aims to conduct a preliminary exploration of FreqPolicy's potential when applied to VLA models. Given that policy learning in multitask environments is still a multifaceted and challenging research problem, we do not aim to provide a solution here. Instead, our goal is to assess FreqPolicy on a multitask benchmark and to provide a forward-looking discussion on whether the core ideas of FreqPolicy, such as frequency-domain decomposition and hierarchical learning, can benefit VLA models. We also outline possible directions for future research. We believe that this discussion can help to provide a more comprehensive understanding of FreqPolicy's potential and its future development.

**Multitask Benchmark.** To preliminarily assess FreqPolicy's performance in a multitask setting, we selected RoboCasa [27] as the benchmark platform. RoboCasa comprises a suite of tasks defined within a simulated kitchen environment, representing complex interaction scenarios that robots might encounter in the real world. In this exploratory experiment, we focused on 24 "atomic" tasks, which cover fundamental sensorimotor skills such as pick-and-place, opening and closing doors, pressing buttons, and turning faucets. Placing our method in such a simulated environment, which possesses a certain level of difficulty and a rich variety of tasks, helps us to preliminarily understand its potential and areas for exploration in multitask learning.

**Baseline.** For effective comparison, we first selected the Diffusion Policy and the autoregressive method BC-Transformer [25] implemented in RoboCasa as direct baselines for our FreqPolicy. Considering the characteristics of VLA models, we further introduced GR00T-N1 [5] as a reference. GR00T-N1 is a VLA model with a tightly coupled dual-system, where its vision-language module is responsible for understanding the environment and instructions, and the subsequent Diffusion Transformer module employs Flow Matching technology to generate smooth action sequences in real-time. Selecting these baselines helps us to more clearly position the relative performance of FreqPolicy within existing VLA frameworks.

**Implementation Details.** In this initial exploration, our FreqPolicy model is directly integrated on top of the Diffusion Policy framework within RoboCasa. To ensure a fair comparison, the main parameters and observation inputs used by FreqPolicy are kept consistent with those of Diffusion Policy. This simplified integration aims to quickly validate the basic adaptability of FreqPolicy's core mechanisms in a multitask scenario, rather than to perform deep customization and optimization.

**Results and Discussion.** The experimental results, as shown in Table 11, offer an initial insight into FreqPolicy's performance on the RoboCasa multitask benchmark. The data indicate that, compared to Diffusion Policy and BC-Transformer, our method demonstrates a certain advantage in overall multitask success rates. This preliminarily suggests that FreqPolicy's frequency-domain processing mechanism, and its hierarchical modeling approach to action sequences, are not only effective for single-task learning but may also bring positive impacts to scenarios requiring the simultaneous handling of multiple tasks. However, when compared to the GR00T-N1 model, which is specifically designed for VLA, FreqPolicy's current performance still shows a gap in multi-task success rates. We attribute this primarily to the fact that FreqPolicy, in its current design and similar to Diffusion Policy, focuses more on action generation and optimization. It has not been specifically enhanced for deep understanding of complex language instructions and multi-modal scene perception to

Table 11: **Multitask results on RoboCasa.** Experimental results of BC-Transformer, Diffusion Policy and GR00T-N1 are from the GR00T-N1 paper.

| | BC-Transformer | Diffusion Policy | GR00T-N1 | **FreqPolicy(Ours)** |
|---|---|---|---|---|
| Success Rate | 26.3% | 25.6% | 32.1% | **27.4%** |

the same extent as GR00T-N1. One of the core strengths of VLA models lies in their powerful semantic understanding and scene perception capabilities, enabling them to generalize better to unseen instruction and environment combinations.

Nevertheless, these preliminary results also provide us with important insights: could FreqPolicy's unique frequency-domain analysis and modeling approach serve as a beneficial supplement when integrated into more powerful VLA frameworks? For instance, FreqPolicy's ability to capture the smoothness and structural information of action signals might assist VLA models in generating more stable and physically plausible action sequences. Exploring how to effectively combine the frequency-domain strengths of FreqPolicy with the semantic understanding capabilities of VLA models, with the aim of further enhancing overall multi-task learning performance, will be a highly valuable research direction for us in the future. This might involve designing new fusion mechanisms or tailoring FreqPolicy's frequency decomposition strategies and autoregressive processes to the specific characteristics of VLA tasks. In summary, while FreqPolicy was not natively designed for VLA tasks, its core ideas demonstrate a potential worthy of further investigation in the broader field of multi-task robot learning.

### F.2 Discussion on Frequency Domain

In the preceding discussions, we have preliminarily shown that the action signals in robotic manipulation tasks exhibit significant compressibility in the frequency domain, with most critical information concentrated in the lower frequency bands. To make this statement precise, we quantify "compressibility" by the energy proportion:

$$E(p) = \sum_{k=0}^{\lfloor (N-1)p\% \rfloor} |X_k|^2 \Big/ \sum_{k=0}^{N-1} |X_k|^2, \tag{4}$$

i.e. the share of total signal energy that is preserved when only the lowest $p\%$ DCT coefficients are retained (see Figure 1).

This section aims to provide a more in-depth discussion and analysis of the frequency-domain characteristics of action signals, based on a broader set of tasks and more detailed visualizations (as shown in Figures 7 to 30). These supplementary figures provide action visualizations (a), frequency band energy heatmaps for each action dimension (b), and success rate curves for actions reconstructed with varying frequency ratios (c) for each task, thereby offering more robust support for our core arguments.

**High-Dimensional Tasks.** In high-dimensional action spaces (22 dimensions) within Dexart tasks (Figures 7-10, Dexart Bucket, Faucet, Laptop and Toilet), we observe consistent trends. The success rate curves (c) for these tasks generally show that even using only 30%-70% of the low-frequency components is often sufficient to reconstruct action sequences capable of task completion, strongly supporting the core hypothesis that high-frequency components contribute relatively little to the macroscopic success of these complex tasks. Concurrently, energy heatmaps (b) clearly demonstrate that different action dimensions exhibit varying dependencies on frequency components; some dimensions (like large-range arm movements) have energy highly concentrated in very low-frequency bands, while others (like fine finger postures) might retain significant energy in relatively higher bands. For instance, in Dexart Faucet (Figure 8b), energy distribution in the 10-30% or even higher frequency bands for some dimensions might correspond to fine adjustments for turning a faucet. Although overall trends are similar, the minimum frequency ratio for high success rates varies slightly across Dexart tasks, with Dexart Laptop (Figure 9c), for example, reaching a success plateau around a 0.4-0.6 frequency ratio, suggesting subtle differences in action signal fidelity requirements for various complex manipulations.

**Low-Dimensional Tasks.** Compared to high-dimensional Dexart tasks, low-dimensional Meta-World tasks (Figures 11-30, typically 4-dimensional action spaces) exhibit a more pronounced

low-frequency dominance. In most Meta-World tasks, success rate curves (c) indicate that only 10%-40% of the low-frequency ratio is sufficient for near-perfect task success, with tasks like Meta-World Coffee-Pull (Figure 13c) and Meta-World Disassemble (Figure 15c) requiring only about 20% low-frequency signal. This suggests higher compressibility in action signals for these simpler robotic tasks, corroborated by their energy heatmaps (b) where most action dimensions show energy concentrated in the lowest 0-10% band, consistent with their typically smoother, direct motion trajectories. Minor exceptions, such as Meta-World Shelf-Place (Figure 28c) or Meta-World Push-Wall (Figure 26c), might need slightly more frequency components due to potentially higher precision demands at the end-effector.

**Discussion.** Synthesizing these analyses, we discover the universality and variability of frequency compression across task types: action signals are generally compressible in both high-dimensional complex and low-dimensional structured tasks, though the required frequency bandwidth varies with task dimensionality, complexity, and operational specifics. The heterogeneity of action dimensions highlighted by heatmap analysis suggests that future work on dimension-adaptive frequency processing, rather than global uniform cutoffs, could be a promising optimization, such as dynamically allocating frequency components based on each dimension's energy to balance performance and efficiency. A deeper understanding of task action frequency characteristics can also guide policy learning algorithm design, for example, by using low-frequency biases or stronger regularization for low-frequency dominated tasks to accelerate learning and enhance generalization.

In conclusion, the additional frequency domain analysis in this section provides more comprehensive empirical support for FreqPolicy's core mechanisms, points to valuable directions for optimizing frequency-based robot learning, deepens our understanding of robotic action nature, and underscores the potential of the frequency-domain perspective in building efficient, robust robotic agents.

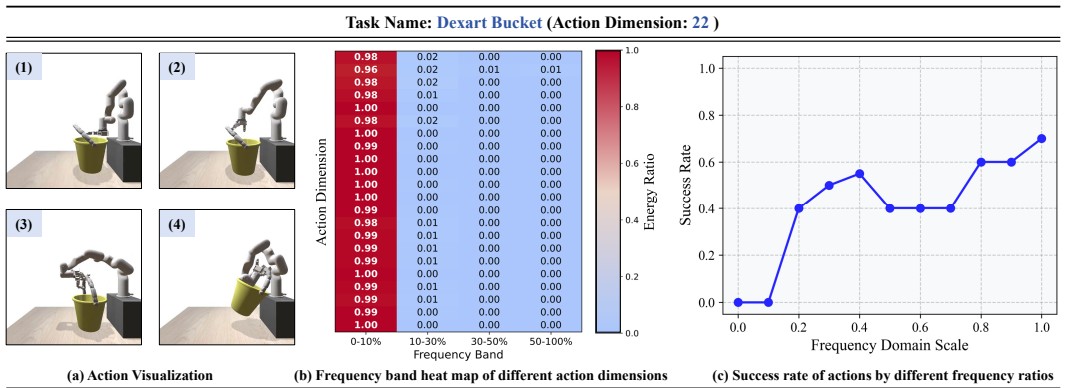

Figure 7: Frequency Domain Analysis of Dexart Bucket.

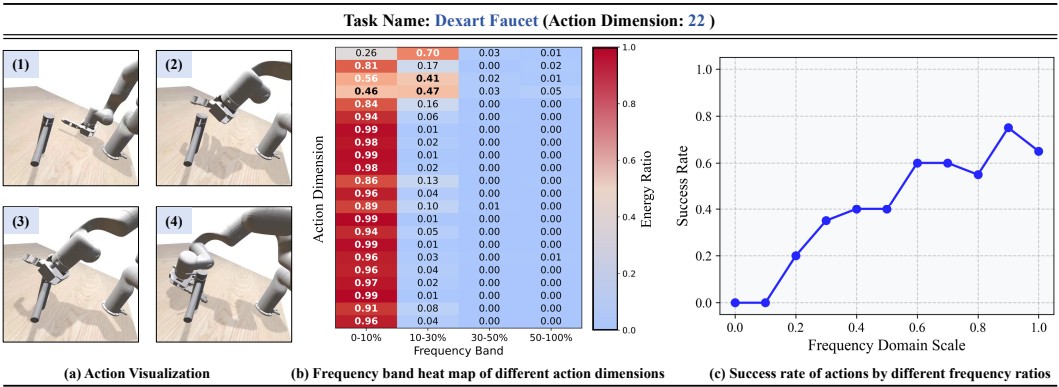

Figure 8: Frequency Domain Analysis of Dexart Faucet.

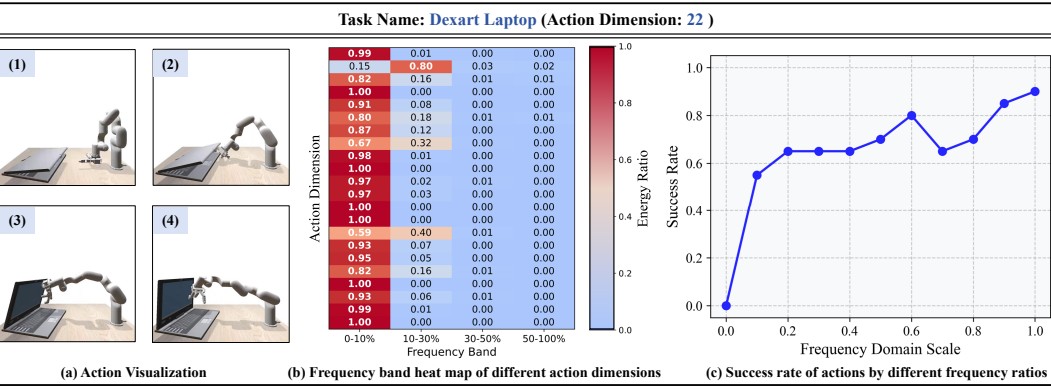

Figure 9: Frequency Domain Analysis of Dexart Laptop.

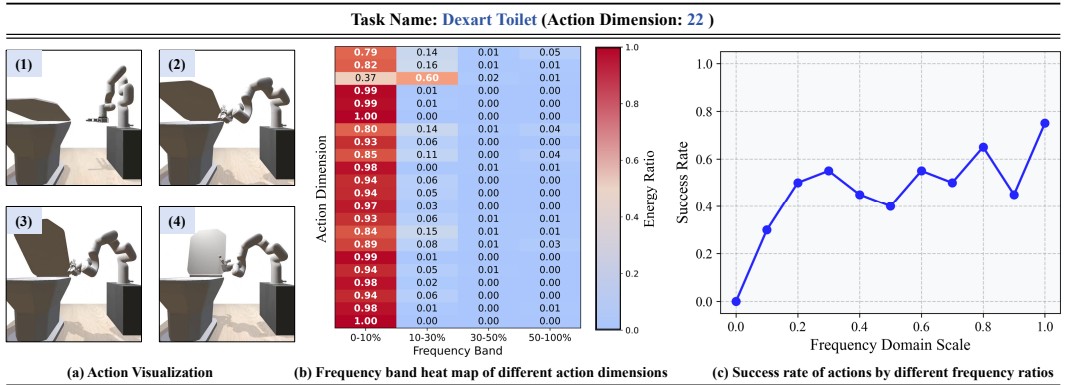

Figure 10: Frequency Domain Analysis of Dexart Toilet.

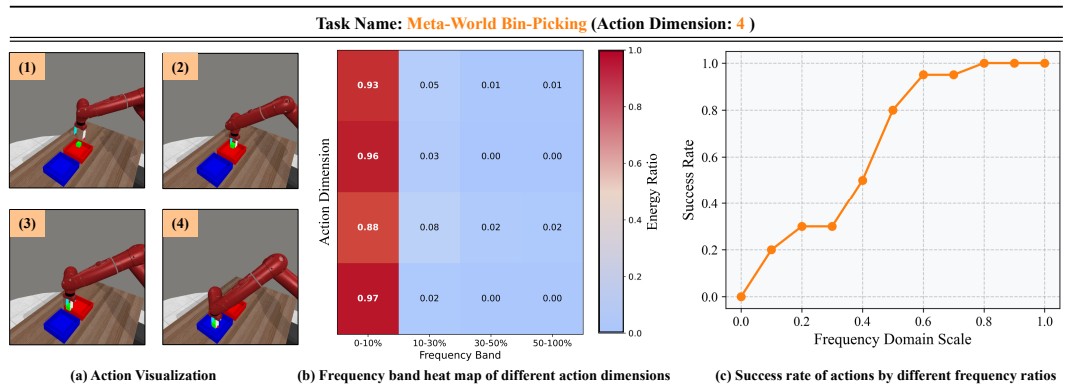

Figure 11: Frequency Domain Analysis of Meta-World Bin-Picking.

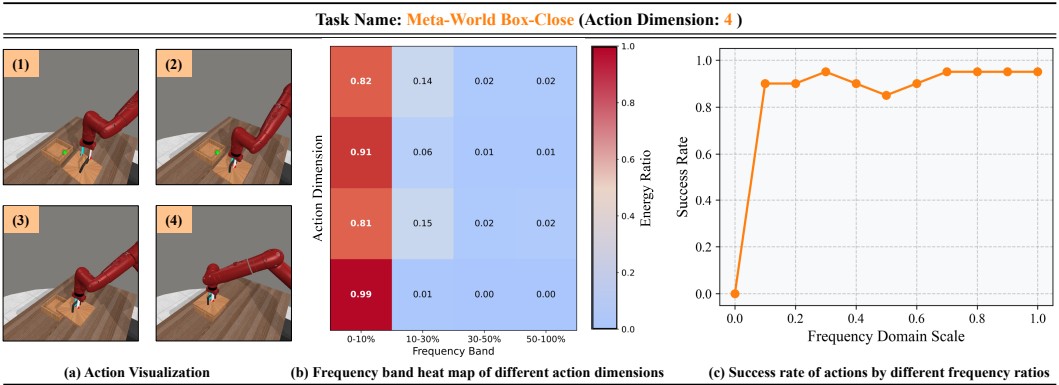

Figure 12: Frequency Domain Analysis of Meta-World Box-Close.

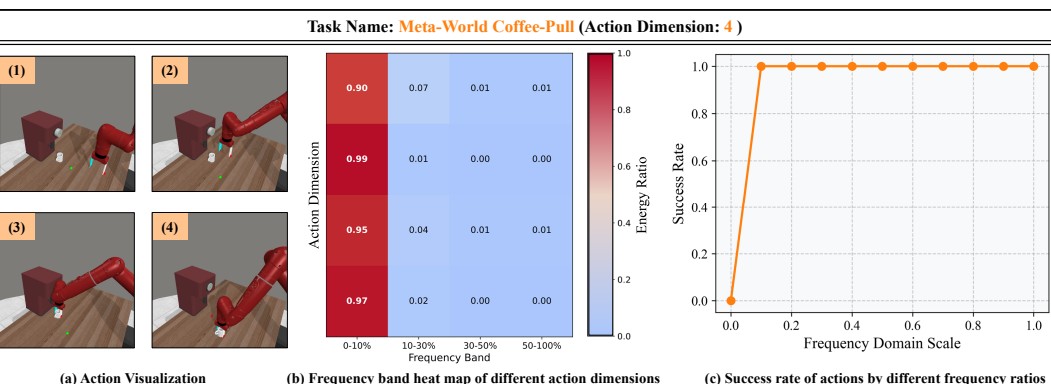

Figure 13: Frequency Domain Analysis of Meta-World Coffee-Pull.

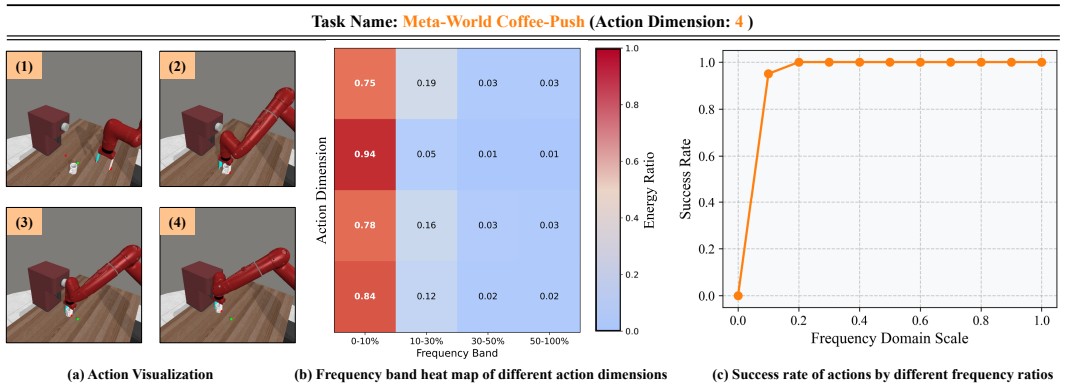

Figure 14: Frequency Domain Analysis of Meta-World Coffee-Push.

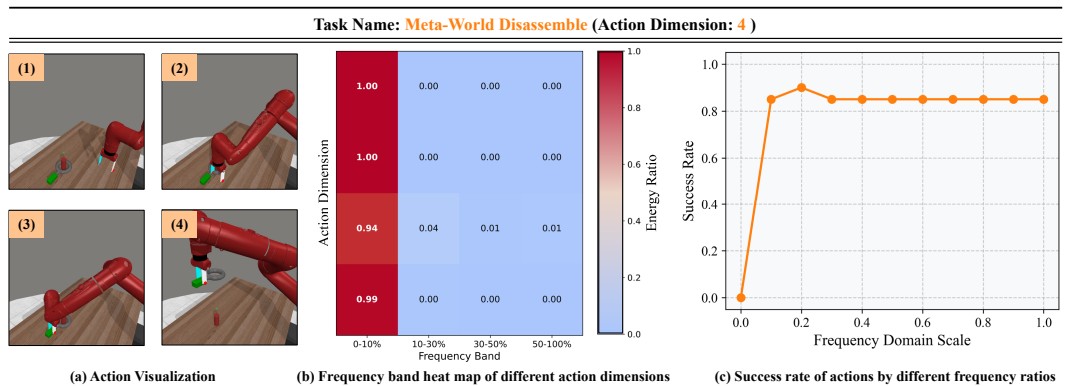

Figure 15: Frequency Domain Analysis of Meta-World Disassemble.

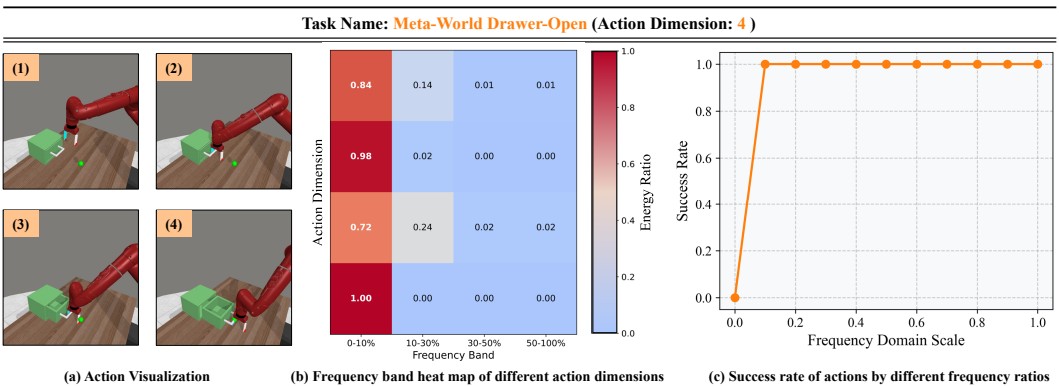

Figure 16: Frequency Domain Analysis of Meta-World Drawer-Open.

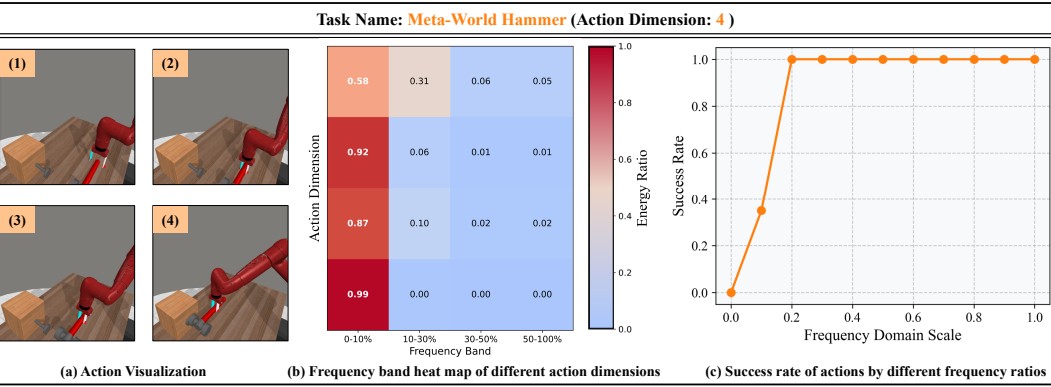

Figure 17: Frequency Domain Analysis of Meta-World Hammer.

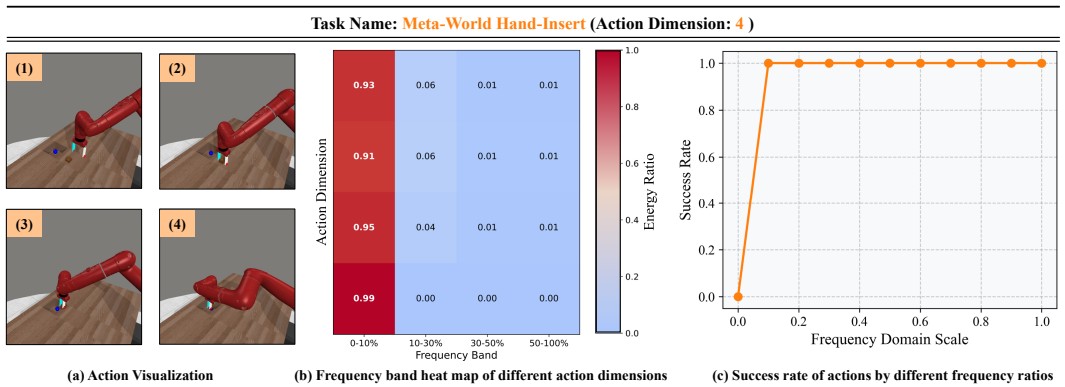

Figure 18: Frequency Domain Analysis of Meta-World Hand-Insert.

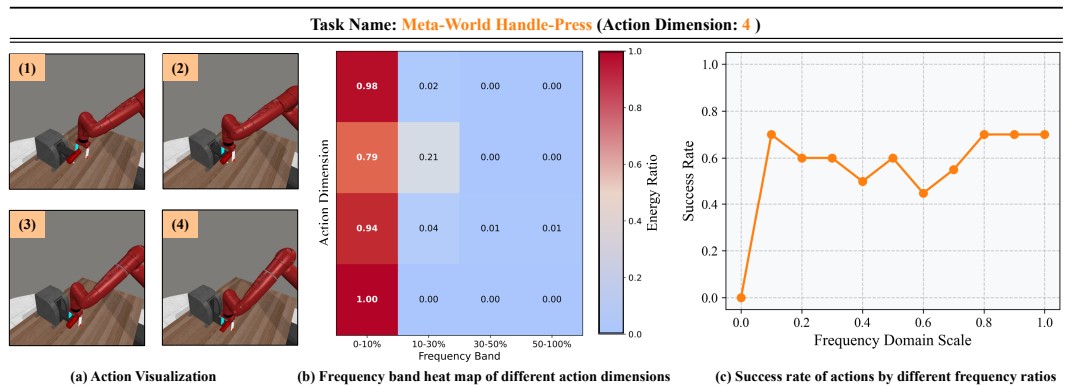

Figure 19: Frequency Domain Analysis of Meta-World Handle-Press.

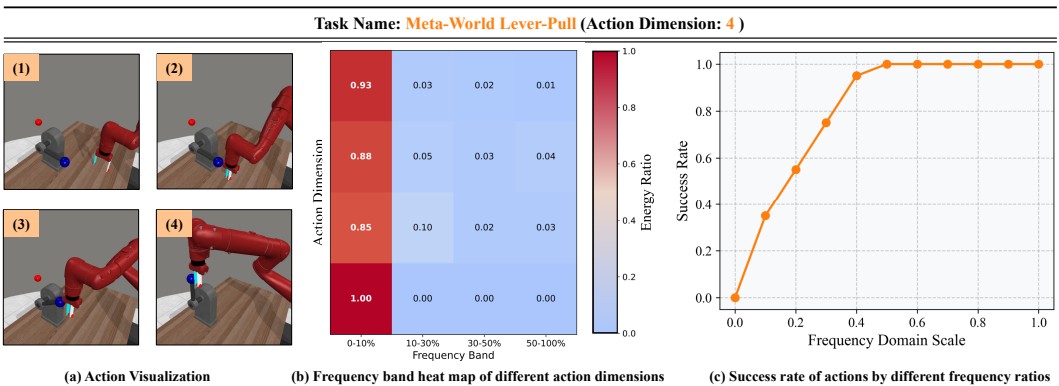

Figure 20: Frequency Domain Analysis of Meta-World Lever-Pull.

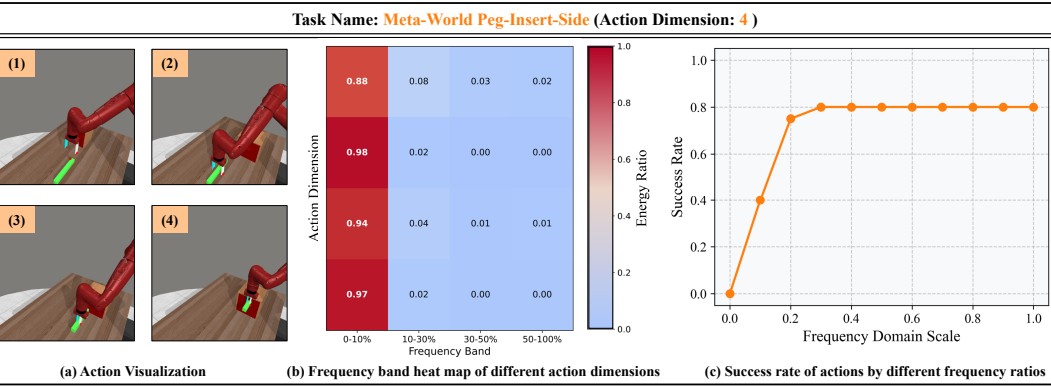

Figure 21: Frequency Domain Analysis of Meta-World Peg-Insert-Side.

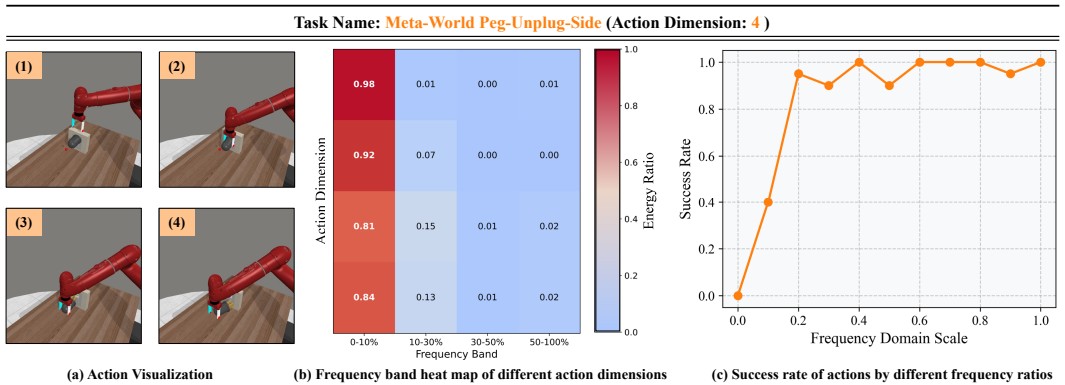

Figure 22: Frequency Domain Analysis of Meta-World Peg-Unplug-Side.

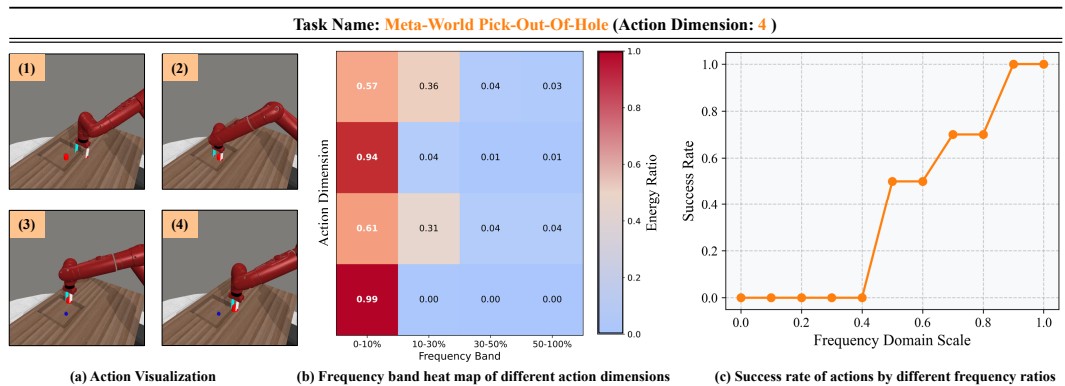

Figure 23: Frequency Domain Analysis of Meta-World Pick-Out-Of-Hole.

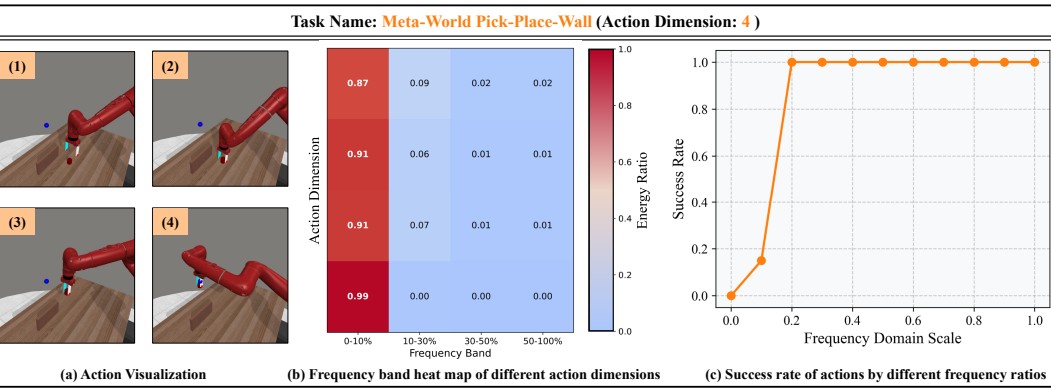

Figure 24: Frequency Domain Analysis of Meta-World Pick-Place-Wall.

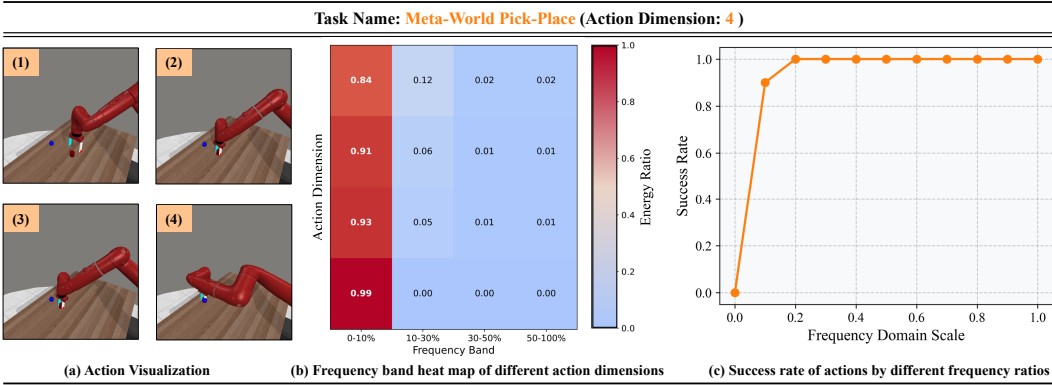

Figure 25: Frequency Domain Analysis of Meta-World Pick-Place.

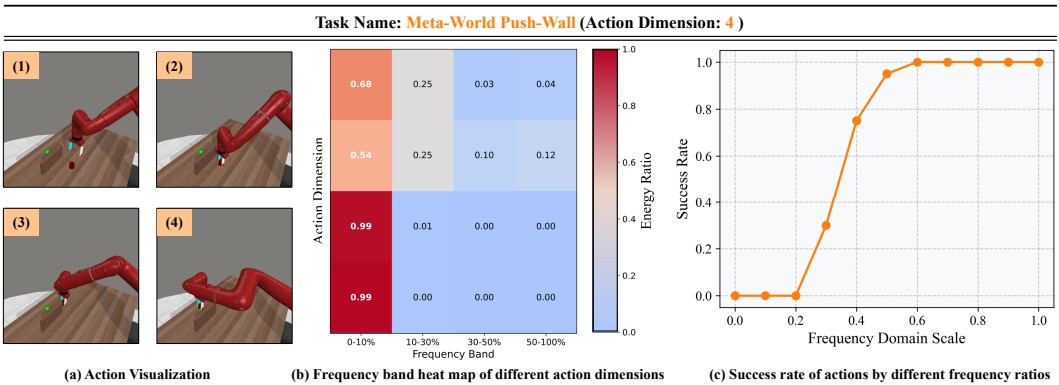

Figure 26: Frequency Domain Analysis of Meta-World Push-Wall.

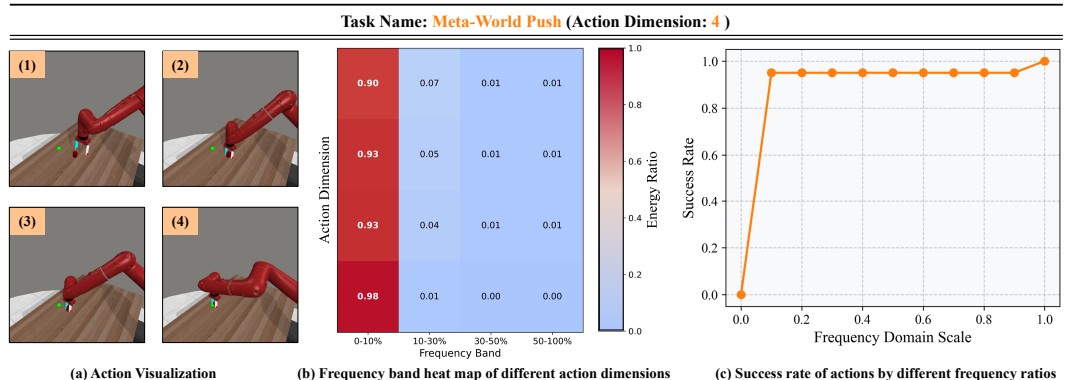

Figure 27: Frequency Domain Analysis of Meta-World Push.

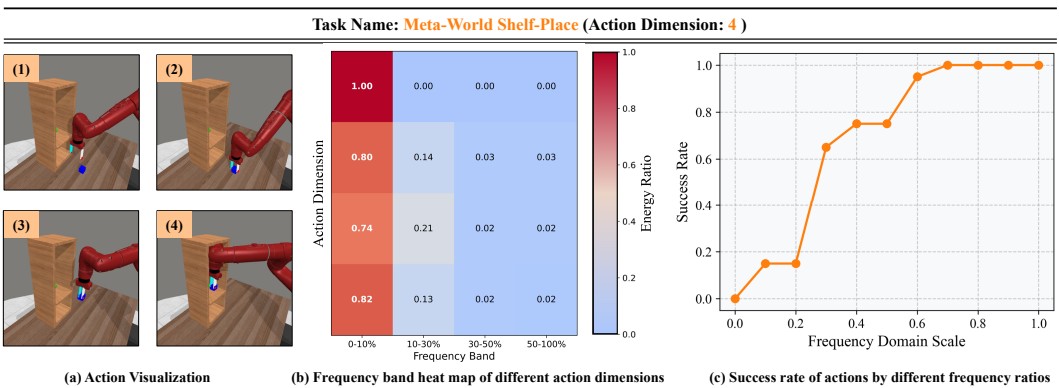

Figure 28: Frequency Domain Analysis of Meta-World Shelf-Place.

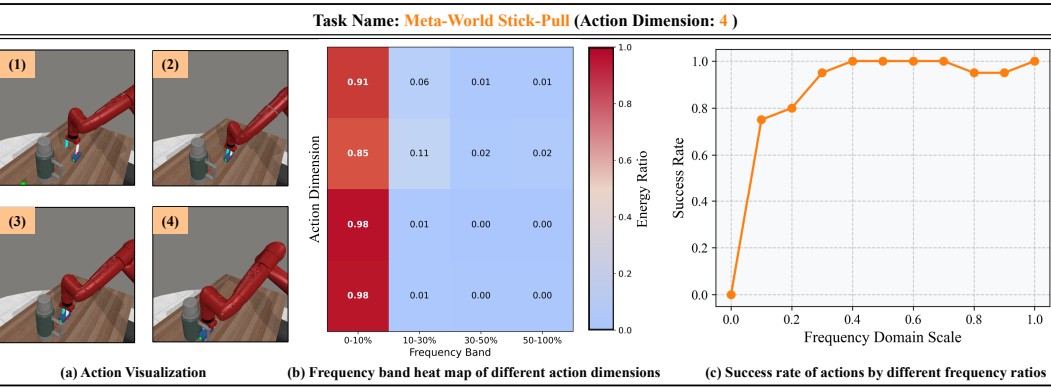

Figure 29: Frequency Domain Analysis of Meta-World Stick-Pull.

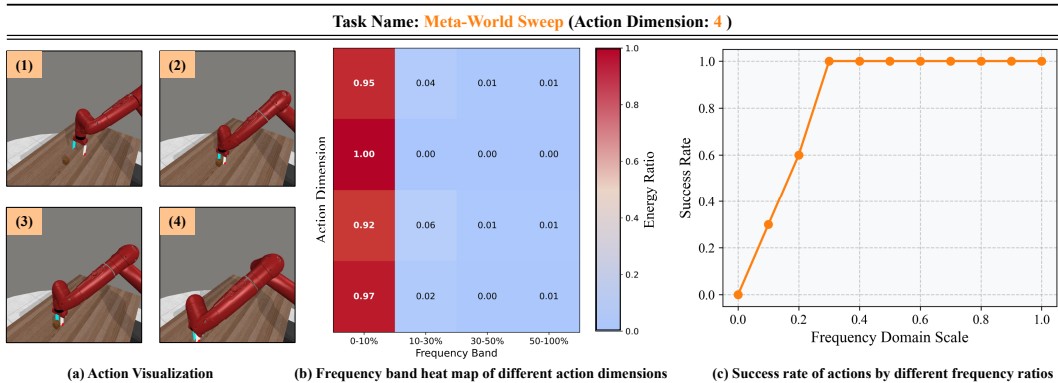

Figure 30: Frequency Domain Analysis of Meta-World Sweep.

