# OpenReview forum: "FreqPolicy: Frequency Autoregressive Visuomotor Policy with Continuous Tokens"
_NeurIPS.cc/2025/Conference — NeurIPS 2025 poster_

### Official Review · Reviewer_Zaqb · 2025-06-02

**Clarity:** 2
**Significance:** 2
**Originality:** 3
**Rating:** 5
**Confidence:** 3

**Summary:**

The paper focuses on an important, often overlooked, aspect of learning robotic policies: *how to represent actions*. Mainly, transforming the actions to the frequency domain provides a certain structure that is effective in identifying global motions (low frequency) and local, fine, motions (high frequency). The proposed action representation and method, FreqPolicy, is composed of hierarchical frequency components (progressively learned), implemented as an autoregressive-diffusion model, and is evaluated on multiple benchmarks resulting in better performance.

**Questions:**

Questions:
* I wrote my questions under “Weaknesses” and “Minor”.

Suggestions:
* I wrote my suggestions under “Weaknesses” and “Minor”.


I'm willing to increase my score given a sufficient response to my concerns (mainly, the clarity issue) and upon reading the other reviews.

**Ethical Concerns:**

["NO or VERY MINOR ethics concerns only"]

**Final Justification:**

My concerns were mostly about the readability and the clarity of the paper. Overall, I like the idea of the paper and I think there is a contribution here to the community, even though the reported results are mostly marginal in compared to other methods (but the method seems to be faster).

**Limitations:**

Very sparse limitations section in the appendix. I can think of some limitations that arise from the complexity involved in synching all the components together (masking and diffusion). I also recommend adding a word or two on this in the main text (e.g., in the Conclusions section).

**Quality:**

3

**Strengths And Weaknesses:**

**Strengths**:
* Well-motivated: the analysis performed on robotic actions and motions justifies the proposed approach.
* Extensive benchmark.
* Faster inference than previous diffusion-based policies.
* Some generalization capabilities (though the overall performance is not great, i.e. <30).
* Real-world demonstration.
* Very detailed appendix.
* A promise for an open-source code.

**Weaknesses**:
* I find the clarity of the method lacking and the entirety of Sections 3.4-3.5 hard to follow. I’m confused regarding the stages and the notations. I would suggest adding a proper example to clarify the sampling/generation pipeline. E.g., “Consider a 3-dimensional action space and a trajectory of 5 timesteps. We apply DCT for $k=...$ levels… For this case, $x_i$ is… $y_i$ is … For hierarchical generation, consider $l_i \in {...}$ and we perform 4 iterations resulting in…”. See “Minor” for more.
* The reported performance improvement seems marginal compared to DP3/Mamba Policy in Tables 1-3
* Misleading results: in Section 4.2 - “Inference Efficiency” - the authors mention that the method is faster compared to diffusion policies, but in Figure 4, to outperform DP3, the method requires 3-4 iterations, which has longer inference time than DP3. I would like a clarification for that.
* The method includes several algorithmic components (e.g., masking, diffusion,...) which might make it a bit complex to implement and tune (though the authors report the hyper-parameters on the appendix).
* No ablations in the main text (only in the appendix). I recommend writing something about it in the main text and short conclusions/insights.
* Limitations: very sparse limitations section in the appendix. I can think of some limitations that arise from the complexity involved in synching all the components together (masking and diffusion). I also recommend adding a word or two on this in the main text (e.g., in the Conclusions section).

**Minor**:
* Introduction - line 44 - you introduce the term “energy” but do not explain its meaning. I also recommend making the caption of Figure 1 a bit more detailed and guide the reader what to look for in the figure (as explained in the main text). In general, this term is used extensively throughout this text and it should be explained in the context of this work or there should be a reference to where it is explained at the very least.
* Figure 2: “We” -> “It”/”The model” (or change all the verbs, i.e, “transforms”, “learns”, “reconstructs”). Also, maybe add numbers to the steps in the figure so the reader can understand where the pipeline starts.
* Please add variable dimensions to section 3.3 (e.g., $x \in \mathbb{R}^{a \times b}$). Also, according to the notation prior to Equation 1, shouldn’t $n$ run from 1 to $N$? There is no $x_0$ in your notation.
* Section 3.5: I’m a bit confused regarding the notations for the index. What is the relationship between $k$ and $l_i$?
* Line 298: “For all simulation experiments, we use 4 iterations, whereas for real-world experiments, only 1 iteration is used”. Iterations of what? Diffusion? Which variable is that referring to?
* Table 4: I assume the numbers are success rates? Please explicitly mention thai in the caption/table.

---

> ### Author Rebuttal · Authors · 2025-07-30
>
> Thank you for your valuable feedback on method clarity, and notation standards. We will implement improvements in the revised version.
>
> **W1. I find the clarity of the method lacking and Sections 3.4-3.5 hard to follow. I'm confused regarding the stages and the notations. I would suggest adding a proper example to clarify the sampling/generation pipeline.**
>
> Here is a detailed example:
>
> - **Detailed Example:** Consider a 3-dimensional action space and a trajectory of 16 timesteps: $x = [x_0, x_1, ..., x_{15}]$ with shape(1×16×3), using 4 iterations with frequency domain indices uniformly divided as index = [0, 4, 8, 12].
>
> - **Training Process:**
>     1. **Random Sampling:** Uniformly sample frequency level k between 0-16, assume we get k = 8
>     2. **DCT Transformation:** $x$ (1×16×3) → $X = [X\_0, X\_1, ..., X\_{15}]$(1×16×3), where $X\_0$ is the lowest frequency component and $X\_{15}$ is the highest frequency component
>     3. **Select First k Components:** Since k=8, select first 8 components $[X_0, X_1, ..., X_7, 0, 0, ..., 0]$ (1×16×3)
>     4. **IDCT Reconstruction:** Obtain 8-level reconstruction $y^8$ (1×16×3)
>     5. **MAE Encoding:** Input ($y^8$ + Obs features + index embedding(k=8)) $\rightarrow$ Output latent $z^8$
>     6. **Diffusion Loss:** Use $z^8$ as condition to compute diffusion loss
>
> - **Sampling Process: (Index embedding omitted in here for brevity)**
>     1. **Iteration 1 (index=0):** Obs features + initial $y^0$ = all zeros(when k=0, reconstructed $y^0$ is also all zeros, perfectly corresponding to the state without any frequency domain information) → MAE encoding → $z^0$ → Diffusion prediction $x^1$ (1st iteration) → DCT, take first 4 components, IDCT reconstruction
>
>     2. **Iteration 2 (index=4):** Obs features + 4-level reconstruction $y^4$ → MAE encoding → $z^4$ → Diffusion prediction $x^2$ (2nd iteration) → Take first 8 components, IDCT reconstruction
>
>     3. **Iteration 3 (index=8):** Obs features + 8-level reconstruction $y^8$ → MAE encoding → $z^8$ → Diffusion prediction $x^3$ (3rd iteration) → Take first 12 components, IDCT reconstruction
>
>     4. **Iteration 4 (index=12):** Obs features + 12-level reconstruction $y^{12}$ → MAE encoding → $z^{12}$ → Diffusion prediction $x^4$ (4th iteration, final full-frequency output)
>
>     This entire process achieves progressive generation from low-frequency to high-frequency components.
>
> **W2. The performance improvement seems marginal compared to DP3/Mamba Policy in Tables 1-3**
>
> We believe our method demonstrates significant advantages across multiple important dimensions:
>
> - **High-DOF Task Advantages:** We achieve significantly higher success rates on high-degree-of-freedom dexterous manipulation tasks. As future demands for high-DOF dexterous operations grow, the advantages of our modeling approach will become more apparent. For example, in Adroit and DexArt benchmarks, our method shows significant advantages on complex manipulation tasks.
>
> - **Flexible Efficiency-Precision Trade-off:** We can efficiently complete tasks under different inference speed requirements by adjusting iteration counts. This adaptability is not available in other methods.
>
> - **Multi-dimensional Performance Improvements:** Beyond success rates, we have significant advantages in: trajectory smoothness, sample efficiency, and noise robustness.
>
> - **Future Potential:** Exploring the combination of autoregressive and diffusion approaches in the action domain has important significance, providing new insights for understanding and modeling action space from different perspectives.
>
> Detailed results can be found in our response to Reviewer wtRU Question 3.
>
> **W3. Misleading results: in Section 4.2 - "Inference Efficiency" - the authors mention that the method is faster compared to diffusion policies, but in Figure 4, to outperform DP3, the method requires 3-4 iterations, which has longer inference time than DP3. I would like a clarification for that.**
>
> Regarding inference efficiency comparison:
>
> - **Comparison with Diffusion Policy:** Our method is significantly faster because Diffusion Policy uses 100 sampling steps, while reducing sampling steps leads to performance degradation.
>
> - **Comparison with DP3:** DP3 uses 10 sampling steps. When we use 3 iterations: (1) inference speed is comparable, and (2) performance is superior to DP3.
>
> - **Experimental Setup:** To ensure fair comparison, we directly use the original paper's values and test according to the same benchmark.
>
> **W4. The method includes several components (masking, diffusion) which might make it complex to implement and tune.**
>
> We made special efforts to simplify the design:
>
> - **Hyperparameter Reuse:** Training hyperparameters such as learning rate directly follow DP3 settings
> - **Diffusion Model Reuse:** Directly use MAR's diffusion model architecture
> - **Standardized Masking Strategy:** Adopt MAR's standard masking configuration, based on open-source MAE model
> - **Lightweight Model:** Only 63M parameters (vs DP3's 255M, see Appendix A), memory consumption ~4.5G, trainable on 2080TI
> - **Open Source Support:** Will open source complete implementation and process documentation
>
> **W5. No ablations in the main text (only in the appendix). I recommend writing something about it in the main text and short conclusions/insights.**
>
> Due to space constraints, we present key ablation study results through Figure 4 in the main text. We will add a brief summary of ablation experiments and main insights in the main text in the revised version.
>
> **W6. Limitations: sparse limitations section in the appendix. Some limitations that arise from the complexity involved in synching all the components together (masking and diffusion). I also recommend adding a word or two on this in the main text (e.g., in the Conclusions section).**
>
> Regarding system complexity, we have addressed this in W4. We will improve the revised version by: Describe the inherent complexity in the main text conclusion.
>
> **M1. Introduction - line 44 - you introduce the term "energy" but do not explain its meaning. I also recommend making the caption of Figure 1 a bit more detailed and guide the reader what to look for in the figure (as explained in the main text). In general, this term is used extensively throughout this text and it should be explained in the context of this work or there should be a reference to where it is explained at the very least.**
>
> Regarding the "energy" concept in Figure 1:
>
> - **Definition:** Energy represents the proportion of signal energy preserved within a specific frequency domain range, calculated as:
> - $E_{total} = \sum |X(k)|^2$ (k = 0 to N-1)
> - $E_p = \sum |X(k)|^2$ (k = 0 to [(N-1)×p%]), where p = 10, 30, 60, 100
> - Energy = $E_p/E_{total}$
>
> - **Role:** As shown in Figure 1, energy proportion reflects the contribution of different frequency domain ranges to signal reconstruction quality. Low-frequency band (0-10%) only preserves 59% energy with high RMSE, while extending to mid-frequency band (0-30%) preserves 95% energy with significantly reduced reconstruction error. This observation provides important insight: adopting the strategy of using the previous frequency domain level as condition for the next level, achieving progressive generation from low-frequency to high-frequency.
>
> We will clearly define the "energy" concept and add complete mathematical formulas in the revised version.
>
> **M2. Figure 2: "We" -> "It"/"The model" (or change all the verbs). Also, maybe add numbers to the steps in the figure so the reader can understand where the pipeline starts.**
>
> We will improve the revised version by:
> - Modify verb forms in Figure 2 to maintain consistency
> - Add numerical markers to process steps in the main text to improve readability
>
> **M3. Please add variable dimensions to section 3.3 (e.g., ). Also, according to the notation prior to Equation 1, shouldn't run from 1 to ? There is no in your notation.**
>
> Thank you for the correction: $x = [x_1, ..., x_N]$ should be $x = [x_0, ..., x_{N-1}]$. We will add variable dimension information in Section 3.3 in the revised version.
>
> **M4. Section 3.5: I'm a bit confused regarding the notations for the index. What is the relationship between k and l_i?**
>
> The relationship between k and $l_i$ is as follows:
> - $l_i$: represents the frequency domain level index corresponding to the i-th iteration
> - In sampling process: $l_i$ is determined according to iteration number i, where $l_i$ is the specific value of k
> - In training process: k is randomly sampled between 0 and N
>
> In short, $l_i$ is the specific instantiated value of k during sampling, while k is the random variable during training. We will further clarify the definitions and relationships of these symbols in the revised version.
>
> **M5. Line 298: "For all simulation experiments, we use 4 iterations, whereas for real-world experiments, only 1 iteration is used". Iterations of what? Diffusion? Which variable is that referring to?**
>
> The "iterations" here refer to the number of iterations in multi-frequency level inference, i.e., the number of intermediate levels in the process of generating actions layer by layer from 0-level to N-level frequency domain, not diffusion steps.
>
> Specifically:
> - **Simulation experiments:** Use 4 iterations, achieving precise action generation through layer-by-layer refinement
> - **Real-world experiments:** Use 1 iteration, directly generating full-frequency domain actions. Since real-world dynamic handover tasks need to respond to rapid environmental changes, single iteration can meet fast inference requirements. Although multiple iterations can provide more fine-grained control, single iteration is more suitable for the high real-time requirements of such dynamic tasks.
>
> **M6. Table 4: I assume the numbers are success rates? Please explicitly mention that in the caption/table.**
>
> Yes, the values are success rates. We will explicitly label "success rate" in the table caption in the revised version.

---

> > ### Comment · Reviewer_Zaqb · 2025-08-01
> > **Thank you for your effort**
> >
> > Dear authors,
> > Thank you for your effort during the rebuttal period. I acknowledge that I have read the other reviews.  Most of my concerns have been adressed and I look forward to read the revised clarified version. Overall, I like the idea of the paper and I think there is a contribution here to the community, even though the reported results are mostly marginal in compared to other methods (but the method seems to be faster). I'm happy to increase my score.

---

### Official Review · Reviewer_FE3t · 2025-06-30

**Clarity:** 2
**Significance:** 2
**Originality:** 2
**Rating:** 4
**Confidence:** 4

**Summary:**

The paper proposes a new method, FreqPolicy, aiming to learn motions progressively with respect to different frequency. FreqPolicy uses a conditioned diffusion model to iteratively generate motions at different levels. The authors compare the method on several simulators to prove its efficiency.

**Questions:**

1.	I think the major problem is the lack of comparison with FAST. FAST also use DCT and decompose the action sequence in the frequency domain. Both the motivation and the method part is very similar. The authors claim that FAST leads to “significant information loss” (L70). It would be definitely helpful if the authors can compare the FAST as a baseline on success rates, reconstruction loss on evaluation set, etc.

2.	For the robomimic benchmark (L166), can you explain the composition of 10 degrees of freedom for robot arm with parallel grippers?

3.	For figure 3(b), what is the x-axis (frequency domain scale)? For frequency domain scale = $1.00$, why the success rates for most of the tasks are still not close to 1.0? How lossy is the reconstruction?

4.	How to represent $y_k$ in the discrete space?

5.	For the experiment results Figure 4, it seems that more iteration (>=4) will lead to a fluctuated success rate (4 > 6 > 7 > 5, etc). If would be better if the author can shed more lights on this. For instance, showing the reconstruction loss of different iterations, etc., will be helpful.

**Ethical Concerns:**

["NO or VERY MINOR ethics concerns only"]

**Final Justification:**

My major concerns are (1) lack comparison with FAST; and (2) definition problem related with DOF. Now the issues have been resolved and I would like to raise the score.

**Quality:**

3

**Strengths And Weaknesses:**

Strengths:

- I really like the visualization and analysis of section 3.2. The motivation of the paper is clear, and is not a pure combination of different methods and tricks, which is insightful.

Weaknesses:

- The writing of the method section is little confusing. It seems that some information in figure 2 is not presenting in the paper, and it took me a while to understand the whole pipeline.

- Comparing with FAST [26] as a baseline seems to be necessary since the similarity of these two methods (both operate in the frequency domain, both use DCT, both try to decompose the actions, but FAST is discrete while FreqPolicy is continuous).

In general, I like this paper, and I'm willing to raise the score if the authors can solve the problems.

---

> ### Author Rebuttal · Authors · 2025-07-30
>
> Thank you for your detailed questions and constructive suggestions. Your feedback from multiple perspectives - method clarity, experimental comparisons, and technical details - has helped us improve the quality of our paper. We sincerely appreciate your thorough review!
>
> **W1.The writing of the method section is little confusing. It seems that some information in figure 2 is not presenting in the paper, and it took me a while to understand the whole pipeline.**
>
> Thank you for the feedback. We have verified that all content and symbols in Figure 2 are mentioned in the method section. To further eliminate confusion, we provide additional clarification of the overall pipeline:
>
> **Core Mechanisms:**
> - **Diffusion Decoding Mechanism:** As shown in Figure 2, FreqPolicy aims to generate actions containing full-frequency information from action tokens at any given frequency level (controlled by index tokens).
>
> - **Specific Training Process:** At index=k, the observation features and corresponding k-level reconstructed actions $y^k$ are input to FreqPolicy's MAE to obtain latent representation $z^k$. $z^k$ serves as the condition for the diffusion model, guiding the generation of full-frequency actions from noise sampling.
>
> - **Hierarchical Frequency Autoregressive Iteration:** The generation process is structured as a for-loop, strictly traversing frequencies from low to high. In the k-th iteration(assuming k iterations divide into k frequency bands):
>   - **Low-frequency extraction:** extracts k-1 level low-frequency actions $y^{(k-1)}$ from the previous prediction's full-frequency action $\hat{x}^{(k-1)}$ through IDCT
>   - **Conditional encoding:** inputs $y^{(k-1)}$ and robot observations O as conditions to FreqPolicy encoder/decoder, obtaining latent embedding $z^{(k-1)}$
>   - **Diffusion generation:** uses $z^{(k-1)}$ as condition input to the diffusion model, generating complete actions $\hat{x}^k$ containing all frequency details
>
> - **Progressive Generation:** At each stage, the model revisits and optimizes the complete action sequence based on increasingly frequency-level and refined actions, enabling progressive generation from coarse-grained to fine-grained.
>
> Concrete examples can be found in our response to Reviewer Zaqb's weakness 1. We will further optimize the clarity of expression in the method section in the revised version.
>
> **Q1.I think the major problem is the lack of comparison with FAST. FAST also use DCT and decompose the action sequence in the frequency domain. Both the motivation and the method part is very similar. The authors claim that FAST leads to "significant information loss" (L70). It would be definitely helpful if the authors can compare the FAST as a baseline on success rates, reconstruction loss on evaluation set, etc.**
>
> Your observation about the missing FAST comparison is very insightful. Following suggestions from both you and Reviewer wtRU, we conducted detailed comparative experiments.
>
> - **Method Route Clarification:** We and FAST adopt two different frequency domain processing approaches:
>   - **FAST Route:** Uses quantization to filter high-frequency components, introducing discretization losses.
>   - **Our Route:** Different tasks need different frequency levels - high-DOF tasks need full-frequency information. We use continuous tokens to preserve action space properties.
>
> - **Experimental Setup:** We tested both FAST+Diff and FAST+AR baseline methods.
>
> - **Experimental Results Comparison (Success %):**
>
>     | Method | Hammer | Door | Pen | Pick Out of Hole | Soccer | Stick Pull | Avg |
>     |:------:|:------:|:----:|:---:|:----------------:|:------:|:----------:|:---:|
>     | FAST+Diff | 13 | 18 | 16 | 32 | 26 | 50 | 26 |
>     | FAST+AR | 0 | 0 | 0 | 0 | 0 | 0 | 0 |
>     | Ours | 100 | 65 | 59 | 30 | 32 | 62 | 58 |
>
> - **FAST Reconstruction Error Analysis (RMSE):**
>
>     | Task | Hammer | Door | Pen | Pick Out of Hole | Soccer | Stick Pull |
>     |:----:|:------:|:----:|:---:|:---------:|:------:|:----------:|
>     | FAST_RMSE | 0.59 | 0.57 | 0.63 | 0.26 | 0.23 | 0.32 |
>
> - **Key Findings:**
>     - **FAST+AR Failure Reason:** Due to frequent token length mismatch issues in FAST's decoder, resulting in all-zero outputs and experimental failure. FAST authors acknowledge this problem in their official repository and suggest using π_0 diffusion decoding as a solution.
>
>     - **Information Loss Verification:** RMSE data clearly validates our claim in Line 70 about "FAST leading to significant information loss." Reconstruction errors in high-DOF tasks (Hammer: 0.59, Door: 0.57, Pen: 0.63) are significantly higher than low-DOF tasks (Pick Out of Hole: 0.26, Soccer: 0.23, Stick pull: 0.32).This explains why FAST achieves reasonable performance on low-DOF tasks but shows degraded success rates on high-DOF tasks.
>
>     - **Performance Difference:** Our method's average success rate (58%) significantly outperforms FAST+Diff (26%), with particularly evident advantages in high-DOF tasks.
>
>     - **Fundamental Method Distinction:** We use full-frequency information with completely lossless reconstruction (guaranteed by DCT's reversible property), while FAST's filtering and quantization operations indeed introduce performance losses.
>
> **Q2.For the robomimic benchmark (L166), can you explain the composition of 10 degrees of freedom for robot arm with parallel grippers?**
>
> The original robomimic dataset consists of eef_pos + eef_rot + gripper = 3+3+1 = 7 DOF. In DP tasks, eef_rot uses 6D rotation representation, resulting in 3+6+1=10 DOF.
>
> **Q3.For figure 3(b), what is the x-axis (frequency domain scale)? For frequency domain scale = 1, why the success rates for most of the tasks are still not close to 1.0? How lossy is the reconstruction?**
>
> - **X-axis Definition:** Frequency domain scale represents the range of frequency domain used (0-100%), i.e., the ratio of k/T. For trajectories with shape B×T×dim, DCT transformation maintains B×T×dim. Using k-level frequency information means adopting B×k×dim.
>
> - **Frequency Information Distribution:** Although trajectory information is mainly concentrated in low frequencies, high-frequency information becomes important as task complexity increases, leading to success rate differences across different frequency domain ranges (see frequency domain analysis videos in supplementary materials).
>
> - **Success Rate Explanation:** When frequency domain scale = 1.0, success rates do not reach 1.0 because:
>    - Testing follows DP/DP3 benchmark protocol, evaluating across 20 random environments
>    - Even RL algorithms that obtain expert demonstrations cannot guarantee 100% success rates across 20 test environments
>    - This reflects task success criteria rather than single-task reconstruction loss
>
> - **Reconstruction Loss:** When using 100% frequency domain, reconstruction loss is necessarily 0, guaranteed by DCT's reversible property. Our method considers all frequency domain information, while FAST's filtering of high-frequency components leads to reconstruction loss.
>
> **Q4.How to represent in the discrete space?**
>
> The representation of $y^k$ in discrete space is achieved through Discrete Cosine Transform (DCT/IDCT):
>
> - **Representation Process:**
>     - **1.Original Trajectory:** Shape B×T×dim
>     - **2.DCT Transformation:** Convert to frequency domain, maintaining shape B×T×dim
>     - **3.Frequency Selection:** Select the first k frequency components, obtaining B×k×dim
>     - **4.Zero Padding:** Set remaining (T-k) high-frequency components to 0, reconstructing B×T×dim
>     - **5.IDCT Reconstruction:** Obtain action space $y^k$ corresponding to k-level frequency domain through inverse DCT
>
> - **Discrete Property Clarification:** The "discrete" here refers to the discrete nature of DCT operations - transformation of finite-length sequences, not numerical quantization. The values in $y^k$ remain continuous floating-point numbers, actually representing k-level reconstruction of the original action sequence, preserving information from the first k frequency components.
>
> This representation method enables us to precisely control information granularity levels in the frequency domain space.
>
> **Q5.For the experiment results Figure 4, it seems that more iteration (>=4) will lead to a fluctuated success rate (4 > 6 > 7 > 5, etc). If would be better if the author can shed more lights on this. For instance, showing the reconstruction loss of different iterations, etc., will be helpful.**
>
> - **Success Rate Fluctuation Reasons:** When iter>4, the following issues exist:
>
>     - **Over-fine High-frequency Division:** High-frequency bands are excessively subdivided, falling into long-tail distribution problems where the model struggles to learn overly subtle high-frequency detail variations
>     - **Noise Interference:** Over-fine subdivision causes the model to learn noise in high-frequency signals, affecting performance
>
> - **Reconstruction Loss Explanation:** Regardless of the number of iterations, when the final frequency level reaches full-frequency domain T level, the reconstruction loss of DCT transformation is always 0. For example:
>     - **iter=1:** Frequency domain divided as 0-100% (full frequency domain)
>     - **iter=2:** Equally spaced division into 50% and 100%, where the 2nd iteration uses 0-50% frequency domain information as condition to generate full-frequency domain actions
>
>     This explains why our method achieves lossless reconstruction while maintaining the flexibility to balance efficiency and precision through iteration control.

---

> > ### Comment · Reviewer_FE3t · 2025-08-04
> >
> > Thank the authors for the response! I have several questions remaining:
> >
> > (1) for Q1, can you further explain the "frequent token length mismatch issues" in FAST's decoder?
> >
> > (2) can you define "high-DOF" tasks and "low-DOF" tasks?
> >
> > (3) for Q2's response, i don't think the system's DOF will be chagned by the rotation representation used. DoF is the property of the physical system itself.
> >
> > (4) for Q3's response, can you further explain the randomness in the environments? let's say you use environment A to collect the data and you use environment B to run the reconstructed action sequence to measure the success rates. is A the same as B or is there some new kind of randomness introduced in B?

---

> > > ### Author Response · Authors · 2025-08-04
> > >
> > > Thank you for your continued engagement! We appreciate your thorough questions and are eager to address each of your concerns.
> > >
> > > **(1) for Q1, can you further explain the "frequent token length mismatch issues" in FAST's decoder?**
> > >
> > > The "frequent token length mismatch issues" stem from FAST's BPE (Byte-Pair Encoding) tokenization approach for discrete frequency representation:
> > >
> > > - **Our Analysis:**
> > >     FAST uses quantization to convert continuous DCT coefficients into discrete tokens, then applies BPE encoding/decoding. However, this process lacks guarantees that the decoded token sequence will match the expected action dimensions. When the autoregressive model generates an incorrect number of tokens, the BPE decoder cannot reshape the coefficients back to the required (time_horizon × action_dim) shape, causing decoding failures.
> > >
> > > - **FAST Team's Confirmation:**
> > >     This issue is acknowledged in FAST's official repository discussions, where users report:
> > >     - **Persistence:** Issues continue even after model convergence (40k training steps)
> > >     - **Root Cause:** BPE tokenizer decoding coefficient matrices that don't match expected shapes (e.g., generating 33 coefficients instead of expected 35 for time_horizon=5, action_dim=7)
> > >     - **Safety Response:** FAST returns all-zero arrays when decoding fails as a "safety mechanism" to prevent potentially dangerous actions from being executed on real robots. However, this conservative approach means the robot remains stationary during decoding failures.
> > >
> > > **(2) can you define "high-DOF" tasks and "low-DOF" tasks?**
> > >
> > > To clarify our classification in the previous responses:
> > >
> > > - **Our Classification:**
> > >     - **High-DOF Tasks:** Adroit benchmark tasks (Hammer, Door, Pen) performed using a 24-DOF ShadowHand
> > >     - **Low-DOF Tasks:** MetaWorld benchmark tasks (Pick Out of Hole, Soccer, Stick Pull) performed using simple parallel grippers
> > >
> > >     This distinction is based on the degrees of freedom of the end-effector rather than task complexity alone.
> > >
> > > - **Standard Definition in Robotics:**
> > >     While there is no universally standardized definition for "high-DOF" vs "low-DOF" tasks in the robotics community, there is a general consensus that dexterous hands with articulated fingers are considered high-DOF manipulators. Therefore, tasks performed using such high-DOF end-effectors can reasonably be categorized as "high-DOF tasks" to distinguish them from simpler gripper-based operations.
> > >
> > > **(3) for Q2's response, i don't think the system's DOF will be chagned by the rotation representation used. DoF is the property of the physical system itself.**
> > >
> > > Thank you for the correction.You are absolutely correct, and we apologize for the confusion in our Q2 response. We made an error in our terminology:
> > >
> > > - **Correction to Q2:** The original robomimic dataset consists of eef_pos + eef_rot + gripper = 3+3+1 = 7 DOF. In DP tasks, eef_rot uses 6D rotation representation, resulting in 3+6+1=10 dimensions.
> > >
> > > - **Clarification:**
> > >     - **Physical DOF:** Always 7 (3 translational + 3 rotational + 1 gripper)
> > >     - **Action Space Dimensionality:** 10 dimensions when using 6D rotation representation (3 position + 6 rotation parameters + 1 gripper)
> > >
> > >
> > > **(4) for Q3's response, can you further explain the randomness in the environments? let's say you use environment A to collect the data and you use environment B to run the reconstructed action sequence to measure the success rates. is A the same as B or is there some new kind of randomness introduced in B?**
> > >
> > > Regarding "Testing follows DP/DP3 benchmark protocol, evaluating across 20 random environments": the randomness here refers to our adherence to the DP/DP3 experimental setup, where we test algorithms using 20 different environment initial conditions within the same task (the differences are mainly in object placement positions). During testing, we do not collect additional data, but only evaluate under these 20 different environment initial conditions.
> > >
> > > Regarding "Even RL algorithms that obtain expert demonstrations cannot guarantee 100% success rates across 20 test environments": this means that when we run the RL checkpoint provided by this benchmark in the aforementioned test environments, it also cannot achieve 100% success rates across these 20 environments. (This RL checkpoint is provided by the benchmark and used to generate expert demonstrations as training data.)
> > >
> > > So in summary, we use the RL checkpoint to generate demonstration data in environment A for training our model, and then the model is evaluated during testing across 20 different initial conditions of the same environment A.

---

> > > > ### Comment · Reviewer_FE3t · 2025-08-07
> > > >
> > > > Thank the authors for the response! I'm happy to see the clarification and new results during the rebuttal. I appreciate the authors' efforts and I'm willing to raise the score.

---

### Official Review · Reviewer_wtRU · 2025-07-02

**Clarity:** 3
**Significance:** 3
**Originality:** 3
**Rating:** 4
**Confidence:** 4

**Summary:**

This paper introduces FreqPolicy, a novel visuomotor policy learning framework that represents robot actions in the frequency domain and generates continuous-action trajectories via a hierarchical, autoregressive diffusion model. By decomposing trajectories into low-to-high frequency components using DCT and mapping them into continuous latent tokens, FreqPolicy progressively refines coarse global motions into fine local details. Extensive simulation results on Adroit, DexArt, Meta-World and RoboTwin benchmarks demonstrate state-of-the-art success rates and a favorable inference–performance trade-off.

**Questions:**

1. Can you provide concrete success‐rate and robustness metrics from your “real-world” experiments rather than only qualitative snapshots?

2. How does FreqPolicy compare against other frequency‐aware action tokenizers such as FAST+DP or DP3 on your core benchmarks?

3. Because of marginal success‐rate gains, what additional evaluation metrics or experiments (e.g., trajectory smoothness in the frequency spectrum, robustness to noise, sample efficiency, or control bandwidth analysis) would you recommend to more convincingly validate the benefits of a frequency-domain approach?

**Ethical Concerns:**

["NO or VERY MINOR ethics concerns only"]

**Final Justification:**

Thank you for the thorough rebuttal. The additional experiments address my previous concerns about validation and significantly strengthen the manuscript. Accordingly, I am happy to raise my overall score.

**Limitations:**

Please see the weaknesses. If you add more experiments to solve my concerns, I would like to raise my scores.

**Quality:**

3

**Strengths And Weaknesses:**

Strengths:

1. Frequency-Domain Representation: Modeling actions via DCT naturally separates global versus fine-grained motion, yielding smoother low-frequency priors and targeted high-frequency refinements—a clear conceptual advance over purely time-domain AR or diffusion approaches.

2. Rich Simulation Evaluation: The authors benchmark on 48 simulation tasks across multiple datasets, outperforming strong baselines (e.g., DP3, Mamba Policy) on 8 of 10 Adroit tasks and averaging 67.9 % success versus 65.8 % from competing methods.

3. Efficiency & Real-Time Potential: FreqPolicy’s single-iteration inference runs at 0.21 s per trajectory (≈ 70 FPS) and demonstrates a qualitative handover on real hardware, highlighting its suitability for latency-sensitive control.

Weaknesses:

1. Lack of Quantitative Real-Robot Metrics: The “real-world” section reports only inference speed and a qualitative handover snapshot, without reporting success rates or robustness on the physical platform.

2. Missing Baseline with Frequency-Tokenizers: Despite the frequency focus, there is no comparison against other frequency-aware action tokenizers such as FAST + DP or DP3 variants that explicitly compress frequency components.

3. Marginal Success-Rate Improvements: Although FreqPolicy outperforms baselines, the absolute gains in success rate are modest (e.g., from 65.8 % to 67.9 % on average), raising questions about the practical significance of the frequency-domain representation.

---

> ### Author Rebuttal · Authors · 2025-07-30
>
> Thank you for your valuable feedback, which helped us significantly improve our experimental validation and comparative analysis.
>
> **W1 or Q1:**
>
> We conducted 10 comparative experiments on real-world dynamic handover tasks with stringent real-time requirements.
>
> - **Experimental Setup:** Dynamic object handover task requires the robot to track and grasp moving objects in real-time, imposing extremely stringent latency requirements on the perception-control loop.
>
> - **Quantitative Results:**
>
>     | Task | DP | DP3 | Ours |
>     |:------:|:--:|:---:|:----:|
>     | Handover | 0 | 0 | 40 |
>
> - **Performance Analysis:**
>   - **DP and DP3:** Due to inference speed limitations (<25 FPS), they cannot meet the real-time requirements of dynamic tasks, resulting in complete failure.
>   - **Our Method:** Using 1 iteration, achieves inference speed of 70 FPS and completes the task with 40% success rate. Demo videos are included in the submitted supplementary materials.
>
> **W2 or Q2:**
>
> - **Method Route Clarification:** We and FAST adopt two different frequency domain processing approaches:
>   - **FAST Route:** Uses quantization to filter high-frequency components, introducing discretization losses.
>   - **Our Route:** Different tasks need different frequency levels - high-DOF tasks need full-frequency information. We use continuous tokens to preserve action space properties.
>
> Following the reviewer's suggestion, we conducted detailed comparisons with FAST+DP3 methods.
>
> - **Experimental Setup:** In addition to FAST+Diff method, we also tried FAST+AR as an experimental baseline.
>
> - **Experimental Results Comparison (Success %):**
>
>     | Method | Hammer | Door | Pen | Pick Out of Hole | Soccer | Stick Pull | Avg |
>     |:------:|:------:|:----:|:---:|:----------------:|:------:|:----------:|:---:|
>     | FAST+Diff | 13 | 18 | 16 | 32 | 26 | 50 | 26 |
>     | FAST+AR | 0 | 0 | 0 | 0 | 0 | 0 | 0 |
>     | Ours | 100 | 65 | 59 | 30 | 32 | 62 | 58 |
>
> - **FAST Reconstruction Error Analysis (RMSE):**
>
>     | Task | Hammer | Door | Pen | Pick Out of Hole | Soccer | Stick Pull |
>     |:----:|:------:|:----:|:---:|:---------:|:------:|:----------:|
>     | FAST_RMSE | 0.59 | 0.57 | 0.63 | 0.26 | 0.23 | 0.32 |
>
> - **Detailed Analysis:**
>   - **FAST+AR Failure Reason:** Due to token length mismatch issues in FAST's decoder, resulting in all-zero outputs. FAST authors acknowledge this problem in their official repository and suggest using π_0 diffusion decoding as a solution.
>
>   - **FAST+Diff Performance Analysis:** From RMSE data, reconstruction errors between FAST-processed actions and ground truth are significantly higher in high-DOF tasks (Hammer: 0.59, Door: 0.57, Pen: 0.63) compared to low-DOF tasks (Pick_hole: 0.26, Soccer: 0.23, Stick pull: 0.32). We believe this is the main factor causing performance degradation due to FAST filtering out partial frequency domains and using quantization operations.
>
>   - **Task Characteristic Impact:** The data further validates our viewpoint that different tasks require different frequency domain levels - high-DOF tasks are more affected by FAST processing, while low-DOF tasks are relatively less affected but still show degradation.
>
> **W3 or Q3:**
>
> **1.Success Rate Analysis:** We achieve significantly higher success rates on high-DOF dexterous manipulation tasks. As future demands for high-DOF dexterous operations grow, the advantages of our modeling approach will become more apparent. For example, in Adroit and DexArt benchmarks, our method shows significant advantages on complex manipulation tasks. Additionally, we can efficiently complete tasks under different inference speed requirements by adjusting iteration counts.
>
> **2.Trajectory Smoothness Analysis:** We evaluate trajectory quality through three metrics:
>
> High-frequency energy ratio (smaller = smoother):
>
> $$\text{ratio} = \frac{1}{D} \sum_{d=1}^{D} \frac{\sum_{f \in F_{\text{high}}} |X_d(f)|^2}{\sum_f |X_d(f)|^2}$$
>
> RMS gradient (smaller = gentler changes):
>
> $$\text{RMS}\_{\text{grad}} = \frac{1}{D} \sum_{d=1}^{D} \sqrt{\frac{1}{N-1} \sum_{t=1}^{N-1} (a_{t+1,d} - a_{t,d})^2}$$
>
> RMS curvature (smaller = more uniform):
>
> $$\text{RMS}\_{\text{curv}} = \frac{1}{D} \sum_{d=1}^{D} \sqrt{\frac{1}{N-2} \sum_{t=1}^{N-2} [(a_{t+2,d} - a_{t+1,d}) - (a_{t+1,d} - a_{t,d})]^2}$$
>
> Where $N$ is time steps, $D$ is action dimension, $a_{t,d}$ is action value, $X_d(f)$ is frequency representation, and $F_{\text{high}}$ denotes high-frequency components.
>
> | Method/Task | Hammer | Door | Pen | Pick Out of Hole | Soccer | Stick Pull | **Average** |
> |:-----------:|:------:|:----:|:---:|:----------------:|:------:|:----------:|:-----------:|
> | **DP3** | 0.067/0.174/0.256 | 0.094/0.202/0.263 | 0.054/0.144/0.187 | 0.118/0.422/0.593 | 0.043/0.079/0.096 | 0.080/0.472/0.681 | 0.076/0.249/0.346 |
> | **Mamba** | 0.066/0.175/0.261 | 0.096/0.209/0.265 | 0.057/0.147/0.188 | 0.097/0.424/0.591 | 0.049/0.089/0.114 | 0.109/0.508/0.721 | 0.079/0.259/0.357 |
> | **Ours** | 0.065/0.171/0.252 | 0.096/0.213/0.267 | 0.051/0.137/0.174 | 0.101/0.390/0.552 | 0.038/0.077/0.093 | 0.076/0.461/0.665 | **0.071/0.242/0.334** |
>
> *Note: Three values in each cell correspond to: high-frequency energy ratio/RMS gradient/RMS curvature*
>
> - **Smoothness Advantage Analysis:** Our method demonstrates advantages in trajectory smoothness due to the progressive generation strategy from low-frequency to high-frequency:
>   - **Low-frequency prior constraints:** In initial stages, the model generates low-frequency information containing main motion patterns and overall trends, naturally possessing smooth characteristics
>   - **Noise filtering mechanism:** Through hierarchical processing, early low-frequency reconstruction effectively filters high-frequency noise interference, providing a more stable foundation for subsequent iterations
>   - **Progressive refinement:** Each iteration refines based on the smooth foundation of the previous level, avoiding sudden changes and discontinuities that might be introduced by direct full-frequency generation
>
> **3.Sample Efficiency:** As shown in Table 4 of the main paper, we achieve significantly better performance than other methods using only 10 demonstration samples.
>
> **4.Noise Robustness:** We conducted two types of noise experiments: 1. Low-quality demonstration experiments using expert demonstrations generated with lowered reward thresholds (5 tasks used, hammer task cannot generate low-reward expert demonstrations) 2. Gaussian noise experiments adding different intensity Gaussian noise to high-quality demonstrations (6 tasks used with 0.025, 0.05, 0.1 std)
>
> -**Noise Robustness Results (Success Rate %):**
>
> ### 1. High-Quality Reference
> | Method | Pick Out of Hole | Soccer | Stick Pull | Hammer | Door | Pen | Average |
> |:------:|:----------------:|:------:|:----------:|:------:|:----:|:---:|:-------:|
> | **DP3** | 12 | 22 | 57 | 100 | 53 | 50 | 49 |
> | **Mamba** | 25 | 28 | 55 | 100 | 59 | 55 | 54 |
> | **Ours** | 30 | 32 | 62 | 100 | 65 | 59 | 58 |
>
> ### 2. Low-Quality Demonstration Experiment
> | Method | Pick Out of Hole | Soccer | Stick Pull | Door | Pen | Average |
> |:------:|:----------------:|:------:|:----------:|:----:|:---:|:-------:|
> | **DP3** | 10 | 15 | 41 | 47 | 33 | 29 |
> | **Mamba** | 16 | 20 | 40 | 44 | 34 | 31 |
> | **Ours** | 20 | 17 | 43 | 49 | 32 | 32 |
>
> ### 3. Gaussian Noise Experiment
>
> **Noise Intensity 0.025 std**
> | Method | Pick Out of Hole | Soccer | Stick Pull | Hammer | Door | Pen | Average |
> |:------:|:----------------:|:------:|:----------:|:------:|:----:|:---:|:-------:|
> | **DP3** | 26 | 25 | 63 | 0 | 55 | 53 | 37 |
> | **Mamba** | 18 | 33 | 61 | 0 | 50 | 57 | 37 |
> | **Ours** | 32 | 33 | 60 | 70 | 52 | 56 | 51 |
>
> **Noise Intensity 0.05 std**
> | Method | Pick Out of Hole | Soccer | Stick Pull | Hammer | Door | Pen | Average |
> |:------:|:----------------:|:------:|:----------:|:------:|:----:|:---:|:-------:|
> | **DP3** | 21 | 19 | 51 | 0 | 27 | 55 | 29 |
> | **Mamba** | 23 | 25 | 60 | 0 | 51 | 57 | 36 |
> | **Ours** | 25 | 30 | 60 | 51 | 46 | 55 | 45 |
>
> **Noise Intensity 0.1 std**
> | Method | Pick Out of Hole | Soccer | Stick Pull | Hammer | Door | Pen | Average |
> |:------:|:----------------:|:------:|:----------:|:------:|:----:|:---:|:-------:|
> | **DP3** | 7 | 18 | 50 | 0 | 24 | 38 | 23 |
> | **Mamba** | 3 | 14 | 39 | 0 | 21 | 39 | 19 |
> | **Ours** | 20 | 27 | 52 | 15 | 32 | 49 | 33 |
>
> **Experimental Results Analysis:**
>
> We discovered the following important phenomena:
>
>   - **Low-quality demonstration impact:** For low-quality demonstrations, all methods show significant performance decline, validating the importance of high-quality training data.
>
>   - **Moderate noise promoting generalization:** Adding lower-intensity Gaussian noise may actually promote generalization ability in 20 random environment tests for certain tasks. For example, in the pen spinning task, Gaussian noise may unintentionally increase hand movement variation, adding action diversity and leading to higher success rates than without noise. Similarly, the Pick Out of Hole task also shows better test performance when low-intensity noise is added.
>
>   - **Hammer task noise sensitivity analysis:** Our method performs particularly well on hammer tasks because hammer tasks require high precision and are particularly sensitive to noise. Our low-frequency to high-frequency progressive generation mechanism has natural anti-noise advantages:
>     - **Low-frequency anti-noise characteristics:** Low-frequency components inherently have stronger noise resistance, establishing a stable action foundation in initial stages
>     - **Progressive filtering:** Each frequency domain level's reconstruction process acts as natural noise filtering
>     - **Structured constraints:** The coarse-to-fine generation process ensures that key action structures are preserved even under noise interference

---

> > ### Comment · Reviewer_wtRU · 2025-08-06
> >
> > Thank you for the thorough rebuttal. The additional experiments address my previous concerns about validation and significantly strengthen the manuscript. Accordingly, I am happy to raise my overall score.

---

### Official Review · Reviewer_ByCW · 2025-07-03

**Clarity:** 2
**Significance:** 3
**Originality:** 2
**Rating:** 5
**Confidence:** 1

**Summary:**

The paper proposes FreqPolicy, a new approach to teaching robots how to move by analyzing actions through their frequency components. The key insight is that robotic movements have natural structure - broad, overall patterns appear in low frequencies while precise, detailed movements show up in high frequencies.
The method works by breaking down actions into these frequency layers and learning them step-by-step in a hierarchical manner. Instead of forcing actions into discrete categories, FreqPolicy employs smooth, continuous representations, combined with a diffusion-based system, to generate more natural and precise movements.
The main innovations include creating a frequency-based framework that captures the inherent structure of how robots move, developing a continuous representation system that avoids the limitations of categorizing actions into fixed bins, and demonstrating superior performance on robot manipulation tasks with better accuracy and faster computation compared to existing approaches.
Essentially, by considering robot actions like how we think about music or signals, with different frequency components conveying distinct types of information, this method achieves more effective and efficient robot learning.

**Questions:**

1.  How does FreqPolicy handle the trade-off between computational efficiency and action precision, especially given that high-frequency components encode fine local details?
2.  What are the specific mechanisms or architectural components within FreqPolicy that enable the "coarse-to-fine" generation process for hierarchical frequency components?
3.  The paper mentions real-world experiments with only 1 iteration for efficiency. How does this single-iteration performance compare to higher iteration counts in simulated environments, and what are the implications for deploying FreqPolicy in other real-time robotic applications?

**Ethical Concerns:**

["NO or VERY MINOR ethics concerns only"]

**Final Justification:**

Thank you for your comprehensive rebuttal. The additional experiments and explanations address my earlier concerns and enhance the quality of the work.

**Limitations:**

Please check the above weakness section

**Quality:**

3

**Strengths And Weaknesses:**

**Strengths:**

  * **Originality:** FreqPolicy introduces a novel and well-motivated approach by representing actions in the frequency domain, addressing limitations of existing methods. The progressive modeling with continuous latent representations is a unique contribution.
  * **Significance:** It significantly advances robotic manipulation by achieving state-of-the-art performance and faster inference, making it highly practical for real-time applications. The analysis of task-dependent frequency requirements is a valuable insight.
  * **Quality:** Extensive experiments across diverse benchmarks consistently demonstrate superior accuracy and computational efficiency. The frequency domain analysis empirically justifies the core idea.
  * **Clarity:** The paper is generally well-written, clearly explaining the problem, method, and results with effective figures.

**Weaknesses:**

  * **Clarity - Specifics on Diffusion-Based Decoding:** More detail on the diffusion-based decoding and its integration in the main text would improve understanding.
  * **Clarity - Masked Generative Strategy:** A slightly more detailed explanation of its benefits (cost reduction, diversity) beyond mask ratios is needed.
  * **Quality - Ablation on Frequency Levels:** The paper lacks explicit discussion or ablation studies on the selection and progression of frequency levels for autoregressive iterations.
  * **Limitations Section:** A dedicated limitations section in the main paper would be beneficial for critical self-reflection.

---

> ### Author Rebuttal · Authors · 2025-07-30
>
> We thank the reviewer for the valuable comments which help improve the paper.
>
> **W1.Clarity - Specifics on Diffusion-Based Decoding: More detail on the diffusion-based decoding and its integration in the main text would improve understanding.**
>
> We will add detailed explanation of the diffusion-based decoding mechanism in the main text:
>
>   - **Diffusion Decoding Mechanism:** As shown in Figure 2, FreqPolicy aims to generate actions containing full-frequency information from action tokens at any given frequency level (controlled by index tokens).
>
>   - **Specific Training Process:** At index=k, the observation features and corresponding k-level reconstructed actions $y^k$ are input to FreqPolicy's MAE to obtain latent representation $z^k$. $z^k$ serves as the condition for the diffusion model, guiding the generation of full-frequency actions from noise sampling.
>
>   - **Iterative Inference:** As illustrated in Figure 2(b), the model starts reconstructing full-frequency actions from low-frequency levels. Through DCT/iDCT operations, it obtains actions containing higher frequency information for the next iteration, achieving progressive action generation from coarse-grained (low-frequency) to fine-grained (high-frequency).
>
> Concrete examples can be found in our response to Reviewer Zaqb's weakness 1.
>
> **W2.Clarity - Masked Generative Strategy: A slightly more detailed explanation of its benefits (cost reduction, diversity) beyond mask ratios is needed.**
>
> We provide additional experimental evidence on the specific benefits of our masking strategy:
>
> | Epoch  | 0 | 400 | 600 | 800 | 1000 | 1200 | 1400 | 1600 | 1800 | 2000 | 2200 | 2400 | 2600 | 2800 |
> |:------:|:-:|:---:|:---:|:---:|:----:|:----:|:----:|:----:|:----:|:----:|:----:|:----:|:----:|:----:|
> | Mask   | 0 | 25  | 42  | 54  | 57   | 65   | 67   | 71   | 74   | 69   | 75   | 73   | 68   | 72   |
> | No Mask| 0 | 4   | 7   | 27  | 30   | 32   | 34   | 42   | 52   | 54   | 64   | 67   | 65   | 71   |
>
>   - **Training Efficiency Improvement:** As shown in the table below, in the average success rate comparison across all tasks in the Adroit benchmark, following DP3 testing protocol with 20 random environments, the masked approach converges faster and achieves better performance:
>
>   - **Diversity Enhancement:** The high average success rates across 20 random environments demonstrate that the masking strategy enhances the model's generalization ability and action diversity across different environments.
>
> **W3.Quality - Ablation on Frequency Levels: The paper lacks explicit discussion or ablation studies on the selection and progression of frequency levels for autoregressive iterations.**
>
> We have conducted ablation studies on frequency level selection for autoregressive iterations in the main text, with results shown in Figure 4. For an action chunk of length T, DCT transformation yields T frequency coefficients. Our key parameter is the number of autoregressive iterations $N_{iter}$, where the frequency spectrum is automatically divided into $N_{iter}$ equal-interval frequency bands, and the model processes these frequency bands sequentially from low-frequency to high-frequency components.
>
> **W4.Limitations Section: A dedicated limitations section in the main paper would be beneficial for critical self-reflection.**
>
> Due to space constraints in the main text, we prioritized presenting the core methodology and experimental results, placing the limitations analysis (detailed in Appendix G) in the appendix. We completely agree with your perspective that this content deserves a more prominent position in the main paper. Therefore, when preparing the final camera-ready version, we will move this discussion to the conclusion section of the main text for clearer self-reflection and future work prospects.
>
> **Q1.How does FreqPolicy handle the trade-off between computational efficiency and action precision, especially given that high-frequency components encode fine local details?**
>
> FreqPolicy handles the trade-off between computational efficiency and action precision directly and flexibly through the number of hierarchical autoregressive iterations $N_{iter}$. Our experimental results (as shown in Figure 4) clearly reveal this: as the number of iterations $N_{iter}$ increases, the model's inference time grows correspondingly, but the task success rate does not necessarily increase. We found that when excessively subdividing and iterating over high-frequency noise, high-frequency data tends to fall into long-tail distributions with less effective signal information, potentially introducing noise instead. In our experiments, $N_{iter}=4$ achieves the optimal balance between performance and efficiency across multiple tasks.
>
> **Q2.What are the specific mechanisms or architectural components within FreqPolicy that enable the "coarse-to-fine" generation process for hierarchical frequency components?**
>
> FreqPolicy's "coarse-to-fine" generation process is achieved through several key architectural components working collaboratively within an iterative autoregressive framework:
>
>   - **DCT/IDCT Transformations:** DCT decomposes action sequences into frequency components, separating "coarse" low-frequency information representing overall motion contours from "fine" high-frequency information for precise local adjustments.
>
>   - **Hierarchical Frequency Autoregressive Iteration:** The generation process is structured as a for-loop (see Appendix Algorithm 2), strictly traversing frequencies from low to high. In the k-th iteration (assuming k iterations divide into k frequency bands):
>     - Low-frequency extraction extracts k-1 level low-frequency actions $y^{(k-1)}$ from the previous prediction's full-frequency action $\hat{x}^{(k-1)}$ through IDCT
>     - Conditional encoding inputs $y^{(k-1)}$ and robot observations O as conditions to FreqPolicy encoder/decoder, obtaining latent embedding $z^{(k-1)}$
>     - Diffusion generation uses $z^{(k-1)}$ as condition input to the diffusion model, generating complete actions $\hat{x}^k$ containing all frequency details
>
>   - **Core Mechanism:** At each stage, the model revisits and optimizes the complete action sequence based on increasingly frequency-level and refined actions, enabling progressive generation from coarse-grained to fine-grained.
>
> **Q3.The paper mentions real-world experiments with only 1 iteration for efficiency. How does this single-iteration performance compare to higher iteration counts in simulated environments, and what are the implications for deploying FreqPolicy in other real-time robotic applications?**
>
> In our real-world dynamic object grasping experiments, we indeed set the iteration number $N_{iter}=1$, primarily determined by the extreme real-time requirements of the task itself. This task involves capturing a moving object, imposing extremely stringent latency requirements on the perception and control loop.
>
>   - **Why Single Iteration:** In this scenario, inference speed is the primary factor determining success. Our FreqPolicy achieves action inference speeds of up to 70 FPS at $N_{iter}=1$, while the baseline method DP3 only reaches 25 FPS. Considering that the perception module also consumes time, only our method can meet such stringent end-to-end latency requirements to successfully complete dynamic grasping. If multiple iterations were used, the FPS drop would prevent the robot from catching up with the moving object.
>
>   - **Consistency with Simulation Results:** This real-world performance is completely consistent with results observed in our simulation environments (as shown in Figure 4). Figure 4 clearly demonstrates that $N_{iter}=1$ corresponds to the highest computational efficiency and shortest inference time configuration. While increasing iteration counts in simulation can further improve success rates for static tasks, single iteration already provides a very strong and efficient performance baseline.
>
>   - **Deployment Flexibility:** FreqPolicy's core value lies in flexibly balancing efficiency and precision according to task requirements:
>     - High real-time tasks: $N_{iter}=1$ maximizes inference speed (e.g., dynamic grasping)
>     - High precision tasks: Appropriately increase iteration counts (e.g., $N_{iter}=4$ for Adroit benchmark)
>     - Real-world static tasks: Can increase iteration counts, FPS drops to 20-30 but success rate improves
>
> This adaptability makes FreqPolicy suitable for a wide range of real-time robotic application scenarios.

---

### Decision · Program_Chairs · 2025-09-17

**Decision:**

Accept (poster)

**Comment:**

The paper proposes FreqPolicy, a frequency-domain autoregressive diffusion policy that learns robot actions from low to high frequencies, and the reviewers agree it is a fresh way to obtain smoother and faster visuomotor control across 2D and 3D benchmarks. Before rebuttal the reviewers were mildly positive, praising the originality and broad experiments but noting hard-to-follow method description, missing baselines, and only modest improvement over strong time-domain policies. After rebuttal the authors gave new FAST comparisons, clearer pipeline examples, and extra real-robot results; the reviewers say these answers solve their concerns and raised ratings to unanimously borderline accept or accept, though limited empirical gains remain a concern. The AC therefore recommends acceptance as a poster.